


# Evaluation of basal melting parameterisations using in situ ocean and melting observations from the Amery Ice Shelf, East Antarctica

Madelaine Rosevear[1,2], Benjamin Galton-Fenzi[3,4,5], and Craig Stevens[6,7]

[1]Oceans Graduate School, University of Western Australia, Perth, Western Australia
[2]Institute of Marine and Antarctic Studies, University of Tasmania, Hobart, Tasmania
[3]Australian Antarctic Division, Kingston, Tasmania, Australia
[4]The Australian Centre for Excellence in Antarctic Science, University of Tasmania, Hobart, Australia
[5]Australian Antarctic Program Partnership, Institute for Marine and Antarctic Studies, University of Tasmania, Hobart, Australia
[6]National Institute of Water and Atmospheric Research, Wellington, New Zealand
[7]Department of Physics, University of Auckland, Auckland, New Zealand

**Correspondence:** Madelaine Rosevear (madi.rosevear@uwa.edu.au)

**Abstract.** Ocean driven melting of Antarctic ice shelves is causing grounded ice to be lost from the Antarctic continent at an accelerating rate. However, the ocean processes governing ice shelf melting are not well understood, contributing to uncertainty in projections of Antarctica's contribution to sea level. Here, we analyse oceanographic data and *in situ* measurements of ice shelf melt collected from an instrumented mooring beneath the centre of the Amery Ice Shelf, East Antarctica. This is the first direct measurement of basal melting from the Amery Ice Shelf, and was made through the novel application of an upwards-facing Acoustic Doppler Current Profiler (ADCP). ADCP data were also used to map a region of the ice base, revealing a steep topographic feature or "scarp" in the ice with vertical and horizontal scales of $\sim$20 m and $\sim$40 m respectively. The annually-averaged ADCP-derived melt rate of $0.51\pm0.18$ m yr$^{-1}$ is consistent with previous modelling results and glaciological estimates, and there is significant seasonal variation in melting with a maximum in May and a minimum in September. Melting is driven by temperatures $\sim 0.2$ °C above the local freezing point and background and tidal currents, which have typical speeds of $\sim$3.0 cm s$^{-1}$ and $\sim$10.0 cm s$^{-1}$ respectively. We use the coincident measurements of ice shelf melt and oceanographic forcing to evaluate parameterisations of ice-ocean interactions, and find that parameterisations in which there is an explicit dependence of the melt rate on current speed beneath the ice tend to overestimate the local melt rate at AM06 by between 200% and 400%, depending on the choice of drag coefficient. A convective parameterisation in which melting is a function of the slope of the ice base is also evaluated and is shown to under-predict melting by 20% at this site. Using available observations from other ice shelves, we show that a common current speed-dependent parameterisation overestimates melting at all but the coldest, most energetic cavity conditions.

## 1 Introduction

The Antarctic Ice Sheet is losing mass, and raising sea level, at an accelerating rate (Bamber et al., 2018; Shepherd et al., 2018; Meredith et al., 2019). This mass loss is caused by the acceleration of the glaciers that make up the Antarctic Ice Sheet


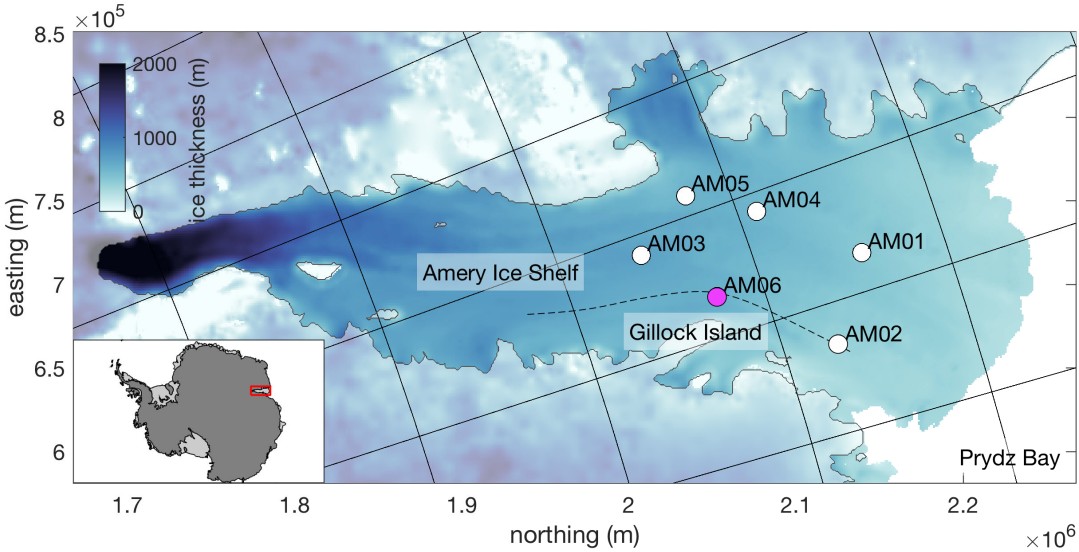

**Figure 1.** Ice thickness map of the Amery Ice Shelf in polar stereographic projection with borehole sites AM01–6 labelled. Site AM06 (magenta) is the focus of this paper. The floating ice shelf is denoted by the bold colours. Map produced with Antarctic Mapping Tools (Greene et al., 2017) using the Bedmap2 product (Fretwell et al., 2013).

in response to reduced buttressing by ice shelves (Dupont and Alley, 2005; Fürst et al., 2016; Reese et al., 2018; Meredith et al., 2019), where the reduction in buttressing is driven primarily by increased sub-ice shelf melting (e.g. Khazendar et al., 2013; Cook et al., 2016; Adusumilli et al., 2018; Minchew et al., 2018). Modelling studies have demonstrated the grounded ice response to enhanced melting is more sensitive in some areas of the ice shelf, such as grounding zones (Reese et al., 2018).

Inter-comparisons of Antarctic Ice Sheet models show that the representation of ocean-induced melting is one of the largest sources of uncertainty in sea level estimates (Seroussi et al., 2020).

Sub-ice shelf melts rates are controlled by ice-ocean interactions involving a range of temporal and spatial scales (Dinniman et al., 2016) from large-scale circulation to micro-scale ice-ocean boundary layer processes (Gayen et al., 2016; Keitzl et al., 2016; Vreugdenhil and Taylor, 2019). Of critical importance to the magnitude, spatial pattern and seasonality of the melt rate

are the properties of the watermasses that intrude into the ice shelf cavity. For example High Salinity Shelf Water (HSSW), a cold ($T \sim -1.9$ °C), dense watermass generated by sea ice formation, tends to drive melting at the back of deep ice shelf cavities. Intrusion of seasonally warmed Antarctic Surface Water (AASW), a much lighter watermass, into cavities drives elevated melt rates in summertime near the ice shelf front (e.g. Arzeno et al., 2014; Stewart et al., 2019). In some locations, relatively warm Circumpolar Deep Water (CDW; $T \sim 1$ °C) drives extremely rapid melting at intermediate depths (e.g. Cook

et al., 2016; Jacobs et al., 2011; Jenkins et al., 2018).





## 1.1 Amery Ice Shelf

The Amery Ice Shelf (AIS) is an embayed ice shelf in East Antarctica with an area of $\sim$62,000 km$^2$ and some of the deepest Antarctic ice in contact with the ocean ($\sim$2200 m; Fricker et al., 2001). Modelling studies suggest that HSSW is present beneath the AIS where it drives moderate melt rates along the eastern flank of the ice shelf cavity (Williams et al., 2001; Galton-Fenzi et al., 2012). The deep draft of the AIS allows HSSW to drive strong melting at the grounding line; a draft of 2200 m depresses the freezing temperature by almost 2°C, resulting in melt-rates that exceed 30 m yr$^{-1}$ (Galton-Fenzi et al., 2012). These elevated melt rates at the grounding line produce cold, fresh, buoyant melt water called Ice Shelf Water (ISW). The ISW ascends the underside of the ice shelf along the western flank of the cavity where it eventually becomes colder than the *in situ* freezing temperature, allowing frazil ice to form and accumulate on the underside of the ice shelf (Fricker et al., 2001; Herraiz-Borreguero et al., 2013). ISW exits the cavity on the western flank of the AIS at depth, creating a cyclonic circulation within the cavity (Herraiz-Borreguero et al., 2016; Williams et al., 2016).

A sustained observational campaign has improved our understanding of circulation in Prydz Bay and beneath the AIS. The Amery Ice Shelf-Ocean Research (AMISOR) project has been monitoring the ocean beneath the AIS for nearly two decades (2001-ongoing) (Allison, 2003). Oceanographic measurements were collected through 6 boreholes from profiling and the deployment of instrumented moorings for longer-term monitoring (Craven et al., 2004, 2014; Herraiz-Borreguero et al., 2013; Post et al., 2014). These moorings confirmed the presence of both HSSW and ISW beneath the ice shelf. Moorings at the AIS calving front observed a modified version of CDW (mCDW) entering the cavity at intermediate depths during the Austral winter (Herraiz-Borreguero et al., 2015), and coincident observations from under-ice mooring AM02 show ISW with a fresher source water mass during this time, suggesting that mCDW drives melting in some areas of the AIS cavity.

A recent remote sensing study estimated the area-averaged melt rate of the AIS to be 0.8±0.7 m yr$^{-1}$ over the period 1994–2018 (Adusumilli et al., 2020), consistent with earlier studies (Yu et al., 2010; Wen et al., 2010; Rignot et al., 2013; Depoorter et al., 2013). Modeling and oceanographic studies report similar values of 0.74 m yr$^{-1}$ (Galton-Fenzi et al., 2012) and 1.0 m yr$^{-1}$ (Herraiz-Borreguero et al., 2016). Galton-Fenzi et al. (2012) showed a seasonal cycle in area-averaged melt with a maximum of 0.8 m yr$^{-1}$ in winter and a minimum of 0.7 m yr$^{-1}$ in summer. The *in situ* data analysed in the present study were collected at site AM06 (Fig. 1), on the Eastern flank of the AIS cavity, and are the first such measurements of basal melting from the AIS.

## 1.2 Melting Parameterisation

The ice shelf-ocean boundary layer (ISOBL) regulates heat and salt exchanges between the ice and the far-field ocean, and plays a crucial role in determining the rate at which the ice shelf melts. The ISOBL can be broken up into two regions: the diffusive sublayer adjacent to the ice, and the turbulent outer layer. In the narrow diffusive sublayer, the effects of viscosity are domiant and heat and salt are transported by molecular diffusion. In the outer layer transport is dominated by turbulent fluxes. The source of this turbulence may be shear instability due to friction between the ocean and ice (e.g. Sirevaag, 2009; Vreugdenhil and Taylor, 2019) or convection due to buoyant meltwater (Kerr and McConnochie, 2015; Keitzl et al., 2016;





Gayen et al., 2016; Mondal et al., 2019; Rosevear et al., 2021; Middleton et al., 2021). An extensive review of the roles of
current shear and convection in ice-ocean interactions can be found in Malyarenko et al. (2020). Notably, they suggest more
work on roughness at all scales should be a future focus of ice-ocean research. The resolution of general circulation models
(e.g. Naughten et al., 2018a) and regional models (e.g. Gwyther et al., 2016; Galton-Fenzi et al., 2012) is far too coarse to
capture the ISOBL processes that regulate melting and a subgrid-scale parameterisation is needed to estimate the melt rate.

Melting is parameterised through a system of equations balancing heat and salt fluxes to the ice-ocean interface with the
latent heat and brine fluxes due to melting (e.g. McPhee et al., 1987; Hellmer and Olbers, 1989; Jenkins, 1991; Holland and
Jenkins, 1999). Further, it is assumed that the interface temperature ($T_b$) is at the freezing temperature at interface salinity ($S_b$)
and pressure ($p_b$, Eq. A1). This results in a system of three equations, which can be solved for $T_b$, $S_b$ and melt rate ($m$). The
way in which oceanic heat and salt fluxes are modelled is the key point of difference between parameterisations.

### 1.2.1   Shear-controlled melting

Shear-dependent parameterisations assume the presence of a turbulent boundary layer formed due to friction between the
stationary ice and the moving ocean. In this type of parameterisation, oceanic heat and salt fluxes are estimated as a function of
friction velocity ($u^*$, a measure of boundary layer turbulence intensity), the bulk temperature and salinity differences across the
boundary layer and turbulent transfer coefficients for heat ($\Gamma_T$) and salt ($\Gamma_T$; Eqs. A2 and A3). It is assumed that this turbulent
boundary layer homogenises temperature and salinity below the ice, forming a well-mixed layer, thus the bulk temperature
and salinity differences are expressed as $T_b - T_{ML}$ and $S_b - S_{ML}$, where $T_{ML}$ and $S_{ML}$ are the mixed layer temperature and
salinity respectively. Friction velocity ($u^*$) is a function of the stress at the ice-ocean interface —which is unresolved in ocean
models—and is typically parameterised as a function of the free-stream current speed ($U$) and an interfacial drag coefficient
($C_d$; Eq. A4).

There are several different expressions in the literature for transfer coefficients $\Gamma_T$ and $\Gamma_S$. Holland and Jenkins (1999),
hereafter HJ99, adopt the transfer coefficients by McPhee et al. (1987) for sea ice melting. McPhee et al. (1987) define $\Gamma_T$ and
$\Gamma_S$ as functions of flow parameters including the Prandtl ($Pr$) and Schmidt ($Sc$) numbers and $u^*$ (Eqs. A5 and A6), as well as
a stability parameter ($\eta$) which describes the stabilising effects of meltwater on the flow. HJ99 showed that for $T' < 0.5\,^\circ$C and
$u^* > 0.1$ cm s$^{-1}$ (corresponding to $U \sim 20$ cm s$^{-1}$) the stability parameter makes less than a 10% difference to the estimated
melt rate. As many of Antarctica's largest ice shelves, such as the Ross, Filchner-Ronne and Amery are thought to be relatively
cold with strong currents, ocean modellers have typically discounted the effects of stabilising buoyancy by setting $\eta = 1$ (e.g.
Galton-Fenzi et al., 2012; Gwyther et al., 2016; Naughten et al., 2018b).

An alternative parameterisation from Jenkins et al. (2010b), hereafter J10, sets the heat and salt transfer coefficients to to
observationally-derived constants $\Gamma_S = 3.1\times10^{-4}$ and $\Gamma_T = 0.011$, assuming $C_d = 0.0097$.

### 1.2.2   Convection-controlled melting

In the convective melting regime, buoyancy —rather than current shear—is responsible for producing turbulence and setting
heat and salt fluxes to the ice. Recent laboratory (McConnochie and Kerr, 2016), turbulence-resolving numerical (Gayen et al.,





2016; Mondal et al., 2019) and theoretical (Kerr and McConnochie, 2015) studies have focused this regime in which melting of sloping or vertical ice is controlled by temperature and does not depend directly on current speed. For a sloping ice-ocean interface McConnochie and Kerr (2018), hereafter MK18, show that $m$ scales as $(T_\infty - T_b)^{4/3}$, where $T_\infty$ is the far-field ocean temperature, and with the basal slope ($\theta$) as $\sin^{2/3}\theta$ (Eqs. A7 & A8). An equivalent expression was determined using turbulence-resolving numerical simulations (Mondal et al., 2019).

A transition from convective to shear-driven melting is expected as flow speeds increase near the ice (McConnochie and Kerr, 2017b). Wells and Worster (2008) proposed a transition based on a critical Reynolds number for the diffusive sublayer ($Re_\delta$), which was applied to observational data by Malyarenko et al. (2020) who found a correlation between $Re_\delta$ and the tendency of either shear or convective parameterisations to reproduce observed melt rates.

### 1.3 Present study

This study presents a set of *in situ* oceanographic and basal melting observations collected beneath the AIS in 2010. Using this unique dataset, we will seek to address the questions: (i) what is the ocean variability in the AIS cavity; (ii) how does this relate to measured melt rate in terms of mean and variation; (iii) how well do available parameterisations predict the basal melt rate in this situation and; (iv) how do these data compare to other published datasets of concurrent melt rate and ocean observations?

## 2 Data and Methodology

The borehole at AM06 (70°14.7' S; 71°28.1' E) was hot water drilled during the 2009/2010 summer at a site that was predicted to be melting (Galton-Fenzi et al., 2012). The ice shelf is 607 m thick, with 73±2 m of freeboard, and the water column thickness is 295 m. The ice-ocean interface and seafloor are at 523±2 and 837±2 dbar respectively.

Several CTD casts were collected over the full depth of the cavity during a two day period using a Falmouth Scientific Instruments (FSI) 3" Micro Conductivity Temperature Depth (CTD) instrument (serial 1610). Pre-season laboratory calibrations of the FSI CTD temperature, pressure and conductivity sensors were done at the Commonwealth Scientific and Industrial Research Organisation Division of Marine Research, however no *in situ* calibrations were performed. Based upon the largest corrections from previous AMISOR sites (using the same instrument) the error is expected to be less than 0.005°C, 0.3 psu and 3 dbar for temperature, salinity and pressure respectively. A mooring, comprising three Seabird SBE37IM Microcats and one upward looking RDI 300kHz Workhorse Acoustic Doppler Current Profiler (ADCP), was then deployed through the ice. All instruments sampled at 30 minute intervals. In the vertical, the ADCP sampled 27 bins at 4 m resolution, where 23 of the bins were within the water column. The beam angles were 20° from the vertical. The location of each instrument with respect to the ice-ocean interface is outlined in Table 1. The duration of the ADCP record is 366 days, thus we restrict the analysis of the Microcat data to the same period for this study. *In situ* temperature ($T$) and Practical Salinity ($S$) are converted to Conservative Temperature ($\Theta$) and Absolute Salinity ($S_A$) using the Gibbs Seawater Matlab package (McDougall and Barker, 2011).



**Table 1.** Type, duration and depth of measurements from the AM06 borehole. Depth given with respect to the ice-ocean interface.

|  | start date | duration (days) | pressure (dbar) | depth (m) |
|---|---|---|---|---|
| *interface* | $\sim$ | $\sim$ | 547 | 0 |
| CTD | 01/01/10 | 2 | 0–837 | 0–286 |
| mCAT1 | 07/01/10 | 366 | 551 | 4 |
| ADCP | 07/01/10 | 366 | 640 | 92 |
| mCAT2 | 07/01/10 | 366 | 681 | 132 |
| mCAT3 | 07/01/10 | 366 | 790 | 286 |

## 2.1 ADCP-derived ice morphology and melting

The range from the ADCP to the ice shelf is used map the ice shelf base and measure the local melt rate. This novel approach used the Bottom Tracking (BT) functionality of the ADCP to map the ice base. A range measurement is obtained from each of the four ADCP beams, which are oriented at $20°$ to the vertical and at $90°$ to each other, and the heading, pitch, and roll data are used to map the range onto a plane in polar coordinates. The instrument rotates about the mooring line due to tidal currents, allowing the ADCP beams to map out a circular swath of the underside of the ice. We find that the BT data are too noisy to recover a direct melt rate measurement. Instead, for the melt rate estimates, range is obtained by post-processing the echo amplitude (intensity) of the ADCP pings. Following the method outlined in Shcherbina et al. (2005), a modified Gaussian is used to approximate the surface reflection peak profile $A(z)$:

$$A(z) = a_0 \exp\left[-\left(\frac{z-h_0}{\delta}\right)^2\right] + a_1 z + a_2 \tag{1}$$

where $a_0$, $a_1$, $a_2$, $h_0$ and $\delta$ are obtained by a least squares fit of Eq. 1 to the echo amplitude data in the vicinity of the surface peak. The fitted value of $h_0$ is an estimate of the range from the ADCP to the ice shelf.

The fit is found for each of the four ADCP beams independently. Ideally, an average over the four beams would be used to decrease the statistical error (Shcherbina et al., 2005). However, this was not possible due to the shape of the ice shelf base (in Sect. 3 we will show that the ice shelf draft changes by 20 m over a horizontal distance of only 40 m), as the surface reflection peak is not captured for ice $\gtrsim 100$ m from the ADCP. Consequently, range is calculated using this method for headings $-30 \leq \theta \leq 46$ only, and melt rate estimates are only possible over this area. All four ADCP beams are mapped onto a plane in polar coordinates and binned in $d\phi \times dr = 4° \times 3$m bins. While the ADCP samples at 30 minute intervals, the sampling frequency for each bin is variable, as it depends on the rotation of the ADCP about the mooring line. Bins are averaged over a month long period to obtain a mean ice-surface position for that month (Fig. B1). Differencing these surfaces yields monthly melt rate estimates.

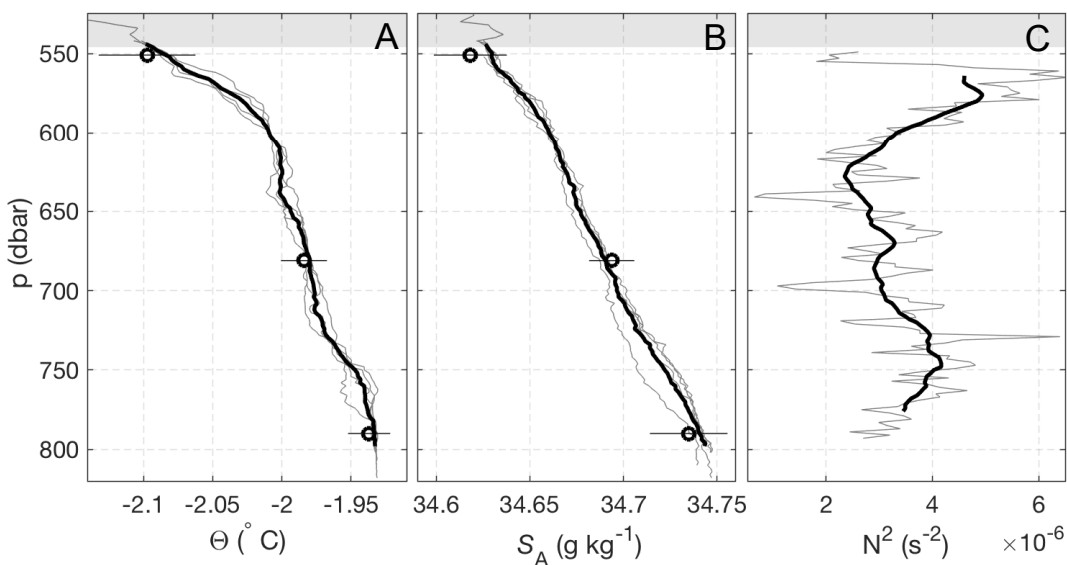

**Figure 2.** (A) Conservative Temperature ($\Theta$) and (B) Absolute Salinity ($S_A$) profiles from the two CTD casts at AM06. Individual up- and down-casts are shown (grey lines) as is the four-profile mean (black line). Overlain at the appropriate pressures are the mean ($\pm\, 2\sigma$) microcat $\Theta$ and $S_A$ for the month following the CTD data collection. (C) Squared buoyancy frequency ($N^2$) profiles, where the grey and black buoyancy frequency curves were obtained using 10 and 40 dbar running window averages respectively. The shaded grey region shows range in the ice-ocean interface position above the instruments due to the sloping ice base.

## 3   Observed hydrography and melting beneath the Amery Ice Shelf

### 3.1   Water column structure

CTD casts collected before the deployment of the mooring at AM06 show a water column stratified in both temperature and salinity, with cooler, fresher water overlying warmer, saltier water (Fig. 2). There is no systematic difference between the up- and down-casts of the CTD, and we include both here. The water column is stably stratified with depth-mean stratification $N^2 \sim 3.5 \times 10^{-6}\ \mathrm{s}^{-2}$, where $N = [-(g/\rho_0)(\partial\rho/\partial z)]^{1/2}$ is the buoyancy frequency. There is no mixed layer beneath the ice, and the temperature gradient is especially strong in the upper 30 m of the water column. There is some evidence for a benthic
mixed layer below $\sim$790 dbar, although this was not consistently sampled by the CTD.

Fig. 3 shows the four-cast mean $\Theta - S_A$ properties from the CTD. A defining feature of ocean conditions beneath ice shelves is the presence of meltwater from meteoric (fresh) ice, which causes ocean properties to evolve along nearly-straight line in $\Theta - S_A$ space (e.g., Gade, 1979; Hattermann et al., 2012; Stevens et al., 2020). Under the assumption of equal eddy diffusivities for heat and salt, the gradient ($d\Theta/dS_A$) of this "meltwater mixing line" can be calculated (Gade, 1979; McDougall et al., 2014).
At local conditions Eq. 16 of McDougall et al. (2014) gives $d\Theta/dS_A = 2.38\ \mathrm{C\ g^{-1}\ kg}$, and explains the $\Theta - S_A$ properties well over the pressure range 560–620 dbar. Below 620 dbar, temperatures remain below the surface freezing temperature,


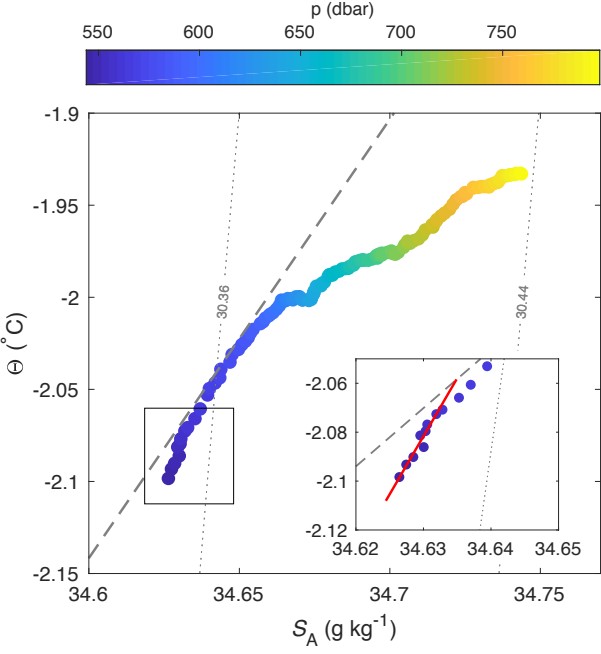

**Figure 3.** $\Theta$-$S_A$ plot of the four-profile mean from Fig. 2 coloured by pressure. A meltwater mixing line with gradient $d\Theta/dS_A$=2.38 °C kg g$^{-1}$ calculated following McDougall et al. (2014) is shown (dashed grey line), as are the local isopycnal slopes (dotted grey lines). The inset shows a line of best fit for the upper 15 m of the water column (red line) which has gradient $d\Theta/dS_A$=4.8 °C kg g$^{-1}$.

however, $\Theta$−$S_A$ properties do not follow a meltwater mixing line, suggesting mixing between two different meltwater-modified watermasses. Below 750 dbar, the HSSW observed is essentially unmodified. Near the interface, the $d\Theta/dS_A$ gradient steepens, deviating from the meltwater mixing line. A line of best fit over this 15 m thick layer has slope $d\Theta/dS_A = 4.8$ C g$^{-1}$ kg. This

result indicates that the turbulent diffusivities of heat and salt are not equal over this region, which could be explained by stratification effects on mixing (Jackson and Rehmann, 2014).

     The timeseries of temperature and salinity measurements from the upper water column indicate that AM06 is a site of melting year-round. For the whole period sampled, the temperature recorded by the upper microcat is greater than the *in situ* freezing temperature at the interface pressure (543 dbar), by ∼0.2 °C (Fig. 5). Temperatures recorded at all depths are colder than the

surface freezing temperature (-1.9 °C) indicating the presence of ISW, and show similar seasonality at all depths (Fig. 5). The water column, which is warmest in summer and autumn, cools over winter and reaches a minimum temperature in spring. The cooling is coincident with freshening at all depths. In October, the cooling and freshening trend reverses, and temperature and salinity increase rapidly towards their previous summer values.

     Microcat temperatures and salinities fall on a melt-freeze line (McDougall et al., 2014), demonstrating that the water masses

arriving at AM06 have been modified by the addition of fresh water due to ice melt, with the highest fraction of melt-water at the microcat nearest the ice base. The ISW present at AM06 follows a single melt-freeze line year round (Fig. 4), suggesting



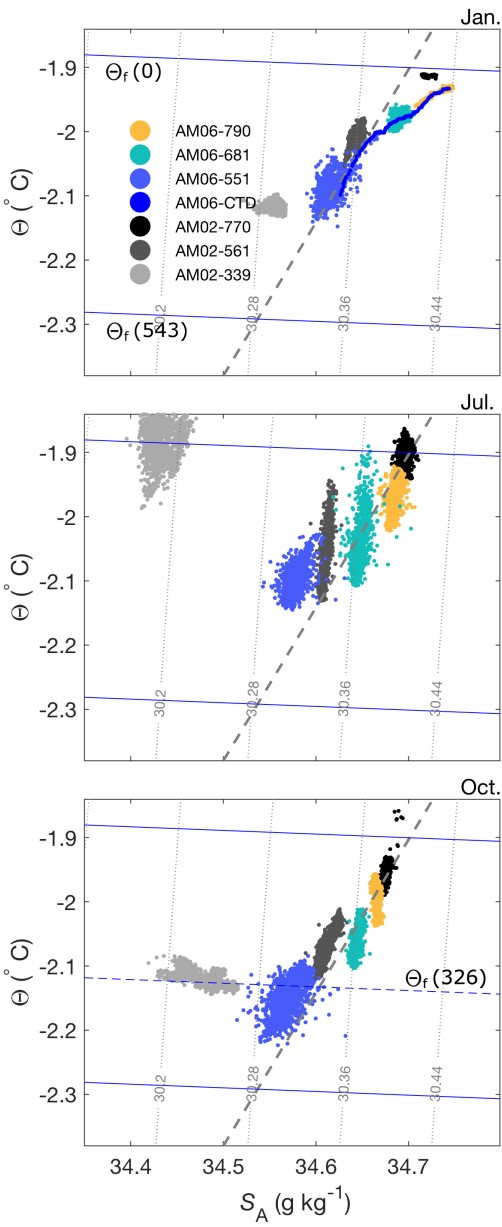

**Figure 4.** $\Theta$-$S_A$ plots for months January, July and October in 2010 at AM06 and AM02 (5 year composite of years 2001, 2003-2006). Freezing temperature curves at surface (0 dbar), AM06 interface (543 dbar) and AM02 interface (326 dbar, bottom panel only) pressure are also shown. The dashed grey line is the meltwater mixing line from Fig. 3.

a single ISW source water mass with $S_A \sim 34.68$ g kg$^{-1}$. Whilst it is not possible to use the melt-freeze relationship to unequivocally identify the source water masses of the ISW, the high salinity suggests that it is HSSW driving melt at AM06,



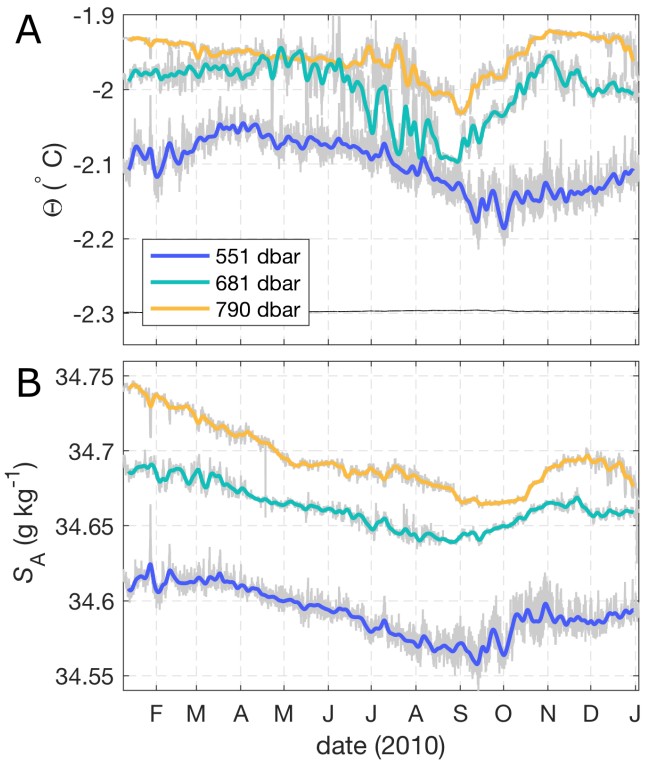

**Figure 5.** (A) Time-series of $\Theta$ and (B) $S_A$ from all three microcats. The freezing temperature (black) at upper microcat salinity and interface pressure is also shown in (A).

where we follow Herraiz-Borreguero et al. (2015) in defining HSSW as having $-1.85 \leq \Theta \leq -1.95$ and $S_A > 34.67$ g kg$^{-1}$.

The $\Theta - S_A$ minimum in spring is likely the result of a high degree of modification of HSSW by meltwater (Fig. 4).

A multi-year composite of $\Theta - S_A$ from mooring AM02 (Herraiz-Borreguero et al., 2015), situated mid-way ($\sim$70 km) from AM06 to the calving front and spanning the years 2001, 2003-2006, is also shown in Fig. 4. The data from AM02 microcats at 561 and 770 dbar show similar seasonality and properties to AM06. At these depths, $\Theta - S_A$ properties at the two mooring locations follow the same meltwater mixing line, suggesting the same source watermass. However, the ocean is

typically cooler and fresher at AM06 than at AM02 at an equivalent depth. This indicates a higher fraction of meltwater at AM06, commensurate with its location deeper in the ice shelf cavity. In July, warmer mCDW is observed at AM02 at 339 dbar (Herraiz-Borreguero et al., 2015), while in October, a watermass at the local freezing temperature with a range of salinities is observed at the same depth. This is likely the result of a rising meltwater plume raised to the *in situ* freezing point by frazil ice formation (Stevens et al., 2020). Temperature and salinity properties at the uppermost microcat at AM02 have no parallel at

AM06, likely due to the deeper ice shelf draft.



**Table 2.** Ellipse parameters for tidal current constituents of the depth-mean flow over the upper 90m of the water column from the T_TIDE harmonic analysis (Pawlowicz et al., 2002) and predicted by the CATS2008 tidal model. The parameters are velocities of the ellipse major and minor axes, inclination of the semi-major axis (counter-clockwise from east), and phase of the tidal vector relative to equilibrium tide at Greenwich.

| | | T_TIDE | | | | CATS | | | |
|---|---|---|---|---|---|---|---|---|---|
| name | frequency | major | minor | inc. | phase | major | minor | inc. | phase |
| | cph | cm s$^{-1}$ | cm s$^{-1}$ | ° | ° | cm s$^{-1}$ | cm s$^{-1}$ | ° | ° |
| S2 | 0.0833 | 3.30 | 0.37 | 69 | 56 | 4.30 | 0.25 | 69 | 57 |
| M2 | 0.0805 | 2.95 | 0.01 | 72 | 313 | 4.18 | 0.24 | 70 | 319 |
| K1 | 0.0418 | 1.79 | -0.01 | 71 | 23 | 2.19 | 0.25 | 73 | 12 |
| O1 | 0.0387 | 1.75 | 0.04 | 72 | 17 | 1.90 | 0.25 | 73 | 0.2 |

## 3.2 Currents

Currents at AM06 have speeds on the order of 10 cm s$^{-1}$ and consist of both tidal and mean components (Fig. 6). To determine the dominant tidal constituents we perform a harmonic analysis of the depth-mean currents using the T_TIDE package (Pawlowicz et al., 2002), restricting the analysis to frequencies higher than semi-annual due to the limited duration of the

ADCP record. The analysis shows that the tides have mixed semidiurnal and diurnal properties. The ellipse semi-major velocities of semidiurnal constituents $S_2$ and $M_2$ are 3.3 and 3.0 cm s$^{-1}$ respectively; larger than than those of diurnal constituents $K_1$ and $O_1$, which are 1.9 and 1.8 cm s$^{-1}$ respectively (Table 2). Ellipse semi-minor velocities for both diurnal and semidiurnal constituents are small, indicating that tides are close to rectilinear. Tides are oriented NNE/SSW, roughly normal to the AIS calving front. At AM06 the CATS2008 circum-Antarctic tide model (an update to the model described in Padman et al., 2002)

performs relatively well against the T_TIDE fits, with the CATS estimates consistently biased 25% high across the constituents examined (Table 2). Our observations of mixed semidiurnal/diurnal tidal currents are similar to previous results from barotropic tidal modelling which showed magnitudes of ~5 cm s$^{-1}$ and ≲2 cm s$^{-1}$ for the semidiurnal and diurnal tides respectively (Hemer et al., 2006).

An estimate of the typical tidal current magnitude is given by:

$$U_{typ} = \sum_{i=1}^{4} (u_{e,i}^2 + v_{e,i}^2)^{1/2} \quad (2)$$

where $u_e$ and $v_e$ are the magnitudes of the semimajor and semiminor axes of the tidal ellipse, with subscript $i$ representing the four main tidal constituents. $U_{typ}$ is roughly the maximum current speed available from these four constituents. Using the constituents in Table 2 yields $U_{typ} = 9.8$ cm s$^{-1}$ at AM06. Instantaneous current speeds at AM06 in excess of 15 cm s$^{-1}$ (Fig. 6) are a consequence of the superposition of mean and tidal currents.


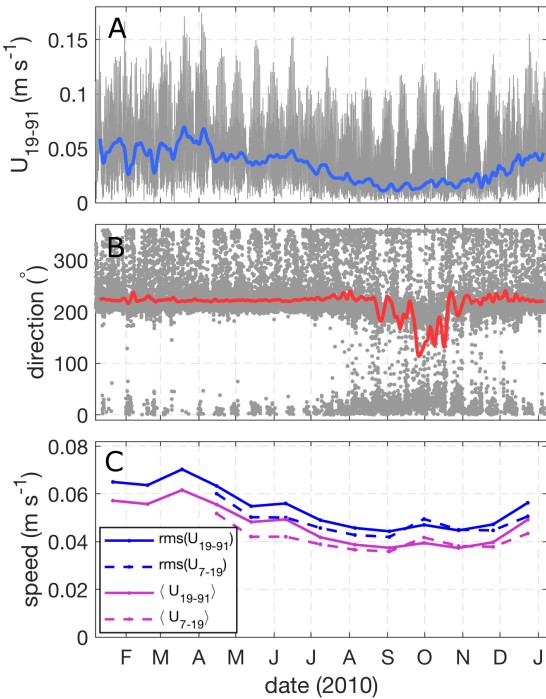

**Figure 6.** (A) Depth-mean current speed $U = \sqrt{u^2 + v^2}$ from 19–91 m below the ice (grey). Overlaid is the smoothed residual current speed ($U_r$, blue). (B) Current direction (grey) and smoothed residual current direction (red). (C) Monthly mean ($\langle U \rangle$) and root-mean-square (rms($U$)) current speed for depth ranges 7–19 and 19–91 m below the ice.

In order to consider the tidal and non-tidal currents separately, we remove the best-fit T_TIDE tidal velocities ($u_T$, $v_T$) from the observed depth-mean velocities, yielding the residual flow $U_r = \sqrt{(u - u_T)^2 + (v - v_T)^2}$. Fig. 6 compares the total and residual flow speeds, where the residual flow is smoothed with a Gaussian filter with a half-width of one week. The residual flow is oriented into the cavity (220°N), has an annual mean speed of 3.2 cm s$^{-1}$ and varies seasonally and at higher frequencies. Notably, in the period August–December $U_r$ is at its weakest and the water column is cold and fresh (Fig. 5), 220  suggesting increased residence time beneath the ice and a higher meltwater fraction at this time. During this period, near-ice current speeds ($U_{7-19}$) are intensified relative to deeper currents ($U_{19-91}$, Fig. 6C).

## 3.3  AIS cavity circulation

The conditions observed beneath the Amery are qualitatively similar to conditions observed in other large cold-type ice shelf cavities, the Ross and Filchner-Ronne, where the water column away from the ice shelf front is dominated by cold ISW (e.g. 225  Nicholls et al., 2004; Stevens et al., 2020). The AIS is the only one of these three ice shelf cavities within which mCDW has been observed (Herraiz-Borreguero et al., 2015). However, the mCDW observed entering the AIS cavity has been heavily modified, and is much colder ($T \lesssim$ -1.6°C) than the CDW in intruding beneath rapidly melting ice shelves such as Pine Island


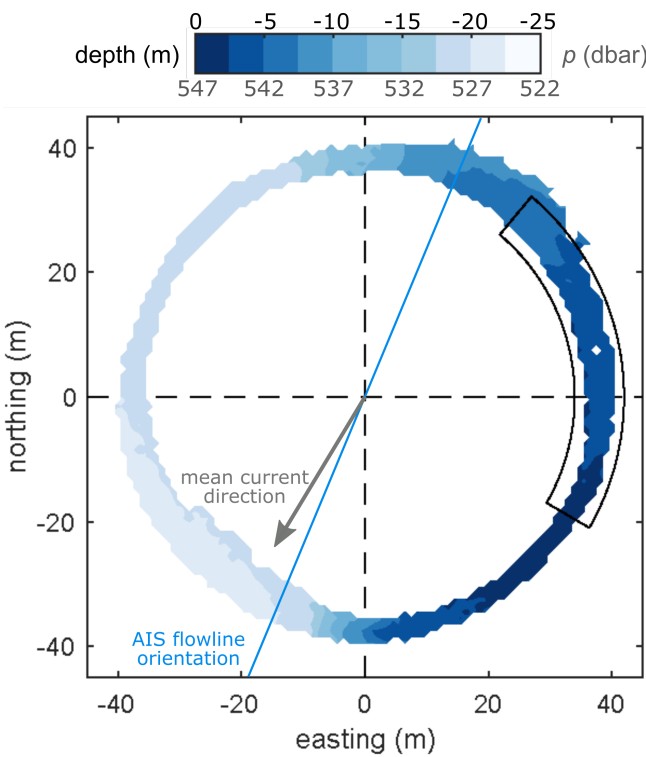

**Figure 7.** The ice shelf base as seen by the upwards-looking ADCP. Zero depth is defined at $p = 547$ dbar, the deepest ice measured at the site, and a negative depth indicates shallower ice. The area mapped out by the ADCP is determined by the beam angle ($20°$) and the distance to the ice shelf (92–114 m). Overlain is an outline of the region of the ice base over which melt rates in Fig 8 are calculated. The local ice shelf flowline orientation and direction of the mean current observed at AM06 are also indicated.

Ice Shelf ($T \sim 1.0°$C, Jenkins et al., 2010a). mCDW was not observed at site AM06, however, the depth of the ice at AM06 ($\sim 600$ m) is likely to exclude mCDW, which only occupies depths $\gtrsim 600$m at the calving front Herraiz-Borreguero et al.

(2016).

Our observations from site AM06 of HSSW-derived ISW with a mean flow of $\sim 3.0$ cm s$^{-1}$ oriented into the cavity supports a three-dimensional model of cavity circulation, consistent with observations from other other AIS moorings (Herraiz-Borreguero et al., 2013, 2015) and modelling results (Galton-Fenzi, 2009). These earlier studies suggested a pathway for HSSW from the calving front along the eastern flank of the AIS, cooling and freshening due to basal melting, with a return

flow on the western flank of the ice shelf driven by a buoyant ISW plume.

### 3.4 ADCP-derived basal melt rate

The ADCP-based approach estimates an annual mean melt rate at AM06 is $0.51 \pm 0.18$ m yr$^{-1}$ (Fig. 8). Monthly-averaged melt rates range from a maximum of 0.8 m yr$^{-1}$ in May to a minimum of 0.2 m yr$^{-1}$ in August and September. The seasonality in

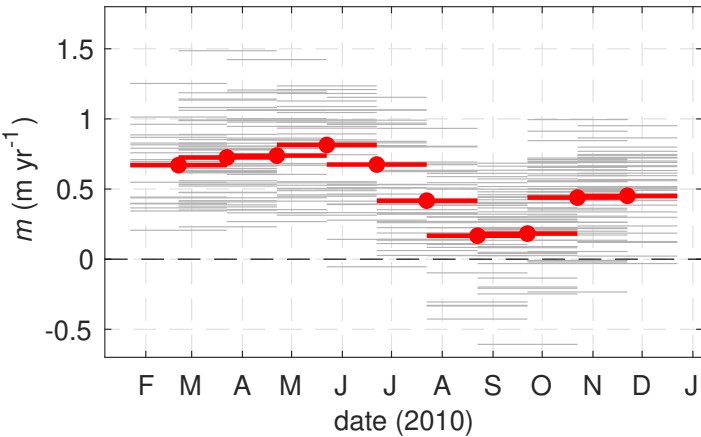

**Figure 8.** Observed melt rates from the ADCP at monthly resolution for (grey) each bin ($d\phi$, $dr$) and (red) the area average, where the area is outlined in Fig. 7.

melting is consistent with the seasonality of upper microcat temperatures. Broadly speaking, when temperatures are warmer,
higher melt rates are observed (Fig. 5). However, the period of maximum melt in June does not coincide with the warmest ocean temperatures, which are observed in April for the upper microcat. Mean and root-mean-squared currents are slightly reduced in the period August–November, coincident with the period of low melt rates (Fig. 6).

The melt rate measured here *in situ* is consistent with the parameterised melt rate in the vicinity of AM06 (Galton-Fenzi et al., 2012, $\sim$1 m yr$^{-1}$), and more broadly with AIS area-averaged melt rates from models (0.74 m yr$^{-1}$; Galton-Fenzi et al.,
2012), oceanographic proxies (1.0 m yr$^{-1}$; Herraiz-Borreguero et al., 2016) and glaciological studies (0.5–0.8 m yr$^{-1}$; Yu et al., 2010; Wen et al., 2010; Rignot et al., 2013; Depoorter et al., 2013; Adusumilli et al., 2020). However, the seasonal cycle in melting at AM06 is somewhat out of phase with the modelled cavity-average, which has a maximum in July and a minimum in January (Galton-Fenzi et al., 2012). The maximum modelled cavity-average meltrate in July leads the period of cooling and freshening at AM06 in July–October, suggesting that the high degree of meltwater modification at AM06 is the result of strong
melting elsewhere in the cavity.

### 3.5  ADCP-derived basal morphology

Data from the ADCP bottom tracking function reveals a large topographic feature or "scarp" in the underside of the ice shelf at AM06 (Fig. 7). The scarp has a vertical extent of $\sim$ 20 m and a maximum slope of 45°. The along-scarp direction is roughly north-south, and the ice deepens moving from east to west. The mean current direction (220°N) is neither along-scarp, as may
be expected if the basal topography was playing an important role in guiding the flow, nor cross-scarp, which could result in flow acceleration, separation and/or blocking. This scarp is an interesting and unexpected feature of the site. There are very few surveys of shelf-undersides at the 1-100 m scale due to difficulty of access. Consequently, the ice base is typically assumed to be smooth, at least at small scales. However, our observation of steep topography with vertical and horizontal scales of $\sim$20





and ∼40 m respectively adds to the small but growing number of studies demonstrating that ice shelves are featured or "rough"
at scales of O(10m) (Nicholls et al., 2006; Dutrieux et al., 2014).

The limited spatial coverage of the ADCP makes is difficult to draw conclusions about the nature or origin of our "scarp",
however, we present several possibilities. For example, the scarp may be an isolated feature, one element of a "rough" patch of
ice (Nicholls et al., 2006) or part of a larger system, for example, a terrace on the flank of a basal channel (Dutrieux et al., 2014)
or a suture zone with its origin upstream. A thorough investigation of the glaciological origins of this scarp is outside the scope
of the present study, however, we note that the scarp is not closely aligned with the local ice flowline (Fig. 7), as would be
expected if the scarp was suture zone between two ice tributaries of different thickness. Nor does it have a surface expression
consistent with a larger channel system. The scarp could also be formed by ocean processes: for example, a convective process
known as double-diffusive convection can drive differential melting of a vertical ice face (Huppert and Turner, 1978). Evidence
of double-diffusive convection has been seen in observations and models of the ocean beneath an ice shelf (Kimura et al., 2015;
Middleton et al., 2021; Rosevear et al., 2021), however, it is likely that the currents at AM06 are too strong for this process to
dominate circulation near the ice and produce such a significant feature through differential melting.

There are many other possible feedbacks between complex topography, flow and melting. For example, acceleration of
buoyant flow up-slope, higher melting on steep slopes (McConnochie and Kerr, 2017b) and differential effects of stratification
on melting for flat vs sloping ice (Vreugdenhil and Taylor, 2019; McConnochie and Kerr, 2016) are all possible. Due to
the background currents, phenomena such as flow blocking, acceleration and separation can also be expected, depending on
ocean stratification and the orientation of the flow with respect to the scarp. These effects will be maximised when the flow is
across-scarp, however, flow at AM06 is primarily along-scarp (Fig. 7). The effects of basal topography on flow and melting
warrant further investigation using observational velocity measurements at high resolution (e.g. sub-meter) and complimentary
modelling studies.

## 4   Evaluation of melting parameterisations

### 4.1   Choice of data

Here we use the concurrent temperature, salinity and velocity measurements from site AM06 to predict the local melt rate
using three different melt-rate parameterisations. We test two current shear-dependent parameterisations solving Eqs. A1, A2
and A3, one with the constant transfer coefficients recommended in J10 and the other using Eqs. A5 and A6 from HJ99. We
also test the convective parameterisation solving Eqs. A1, A7 and A8 from MK18. The mixed layer temperature ($T_{ML}$) and
salinity ($S_{ML}$) are taken from the upper microcat. Because of the sloped ice-ocean interface, the positioning of this instrument
with respect to the ice depends on the region of ice considered. Here we use the lower bound on the interface position ($z$=-541
m) as our reference depth. The upper microcat is therefore situated at a depth of 4 m (Table 1).

All parameterisations included in this study assume a water column structure where strong mixing in a boundary current or
plume produces a well-mixed layer of water near the ice such that, so long as they are measured within the layer, temperature
and salinity do not vary in depth. However, the CTD profiles collected at the start of the observational period do not convinc-





ingly demonstrate the presence of a mixed layer adjacent to the ice (Fig. 2). As such, our results will be sensitive to the depth at which $T_{ML}$ and $S_{ML}$ are taken. Over the upper 10 m of the water column the temperature gradient is $d\Theta/dz = 0.0017\,°\mathrm{C}$ $\mathrm{m}^{-1}$. We assess the sensitivity of the predicted melt rate to the depth at which the temperature is taken in Sect. 4.3.

In the melting parameterisations, Eqs. A2, A3, A5 and A6 require the friction velocity ($u^*$) to estimate turbulent heat and salt transport to the ice. The Law of the Wall (LOW) relates $u^*$ to the near-ice velocity structure via the logarithmic expression $u(z) = (u^*/\kappa)\ln(z/z_0)$, where $\kappa = 0.41$ is Von-Karman's constant and $z_0$ is the roughness length. The velocity profiles recorded by the ADCP were analysed for a logarithmic profile, however the vertical resolution proved to be insufficient to capture the log layer. As such, in Eq. A5 we model $u^*$ using A4 with $C_d = 0.0025$, for which the flow speed ($U$) outside

of the log layer is needed. Typically, the log layer occupies $\sim$10% of the total boundary layer depth, and for the polar oceans is typically in the range 2–4 m deep (McPhee, 2008). The Ekman layer depth scales as $\delta \sim u^*/|f|$, where $f$ is the Coriolis frequency. For a free stream velocity $U = 5.0\,\mathrm{cm\,s}^{-1}$, $f = -1.4\times10^{-4}\,\mathrm{s}^{-1}$ and using Eq. A4 with $C_d = 0.0025$ we find $\delta \sim 18$ m and therefore a log layer depth of $\sim$1.8 m. The uppermost bin sampled by the ADCP is 3 m below the ice, however, this data point is likely to be contaminated as it falls within the upper 6% of the instrument range (Teledyne, 2006). This point has

therefore been discarded. As the upper water column velocity structure is relatively homogeneous in depth we use the four-bin mean over 7–19 m ($U_{7-19}$). This averaging increases data return, and does not bias the speed low or high compared to taking the velocity at 7 m only. There is a significant amount of missing velocity data in the upper water column from January through to early April. As such, the shear-dependent parameterisations are tested using data from April onwards. To test the constant transfer coefficient parameterisation from J10 we take the recommended values $\sqrt{C_d}\Gamma_T = 0.0011$ and $\sqrt{C_d}\Gamma_S = 3.1\times10^{-5}$.

Finally, to apply the MK18 parameterisation, the basal slope of the ice ($\theta$) is needed. As the melt rate was measured over the section of ice bounded by $-30° \leq \phi \leq 46°$, $\theta$ is taken to be the average basal slope within this region, $\theta = 9°$ (Fig. 7). It is worth noting that this is a large slope with respect to the overall ice shelf slope; the mean slope of this ice shelf will be of the order $\tan^{-1}(H/L) = 0.2°$, where $H$ ($\sim$2 km) is the approximate thickness change and $L$ ($\sim$600 km) is the approximate length of the AIS.

**4.2    Ice shelf heat flux**

At the ice-ocean interface, the heat flux from the ocean is balanced by latent heat loss due to melting and conductive heat loss to the ice shelf (Eq. A2). In order to estimate the ice shelf heat flux, we model heat transport within the ice shelf as a balance between vertical advection and diffusion. We assume that the vertical velocity is equal to the basal melt rate and constant within the ice shelf and that the ice shelf is in a steady state, meaning all ice removed from the base is balanced

by surface accumulation (for a thorough discussion around different ice shelf heat transport approximations see HJ99). The advection-diffusion balance is given by:

$$\frac{\partial^2 T_i}{\partial z^2} + \frac{m}{\kappa_{T,i}}\frac{\partial T_i}{\partial z} = 0. \tag{3}$$

We can solve Eq. 3 over an ice shelf of thickness $H$ with surface temperature $T_s$ and basal temperature $T_b$. At AM06 $H = 607$ m, $T_s \sim -20\,°\mathrm{C}$ and $T_b \sim -2.1\,°\mathrm{C}$. At the annual average melt rate of 0.51 m yr$^{-1}$ this model gives a heat flux into the ice of





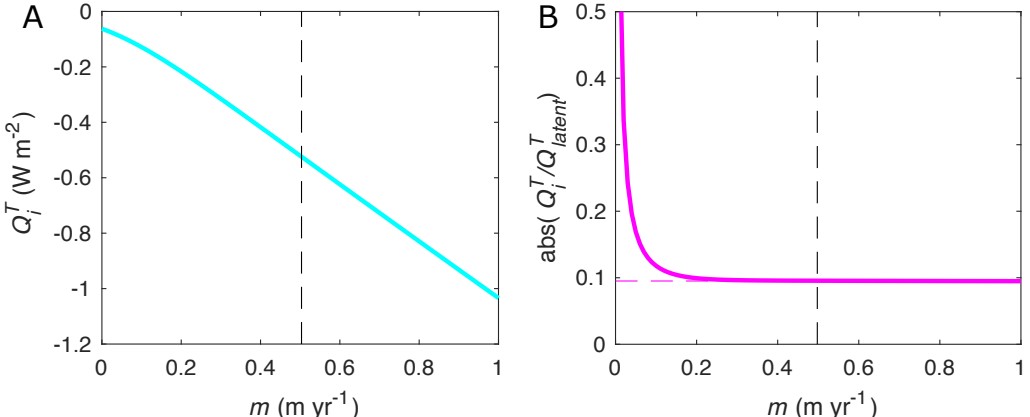

**Figure 9.** (A) Heat flux into the ice ($Q_i^T$) as a function of melt rate $m$ from a one-dimensional advection-diffusion model for a 607 m thick ice shelf with a $-20\,^\circ$C surface and $-2.1\,^\circ$C basal temperature. (B) Ratio of $Q_i^T$ to latent heat flux due to melting $Q_{latent}^T$ as function of $m$. The vertical dashed lines correspond to the annual average melt rate of 0.51 m yr$^{-1}$.

**Table 3.** Melt rates from observations and parameterisations. Columns 2–4 are average values observed over the periods given in column 1, where the period starts on the 7th day of the month. Current speed $U$ is depth-averaged over 7–19 m below the ice base. To calculate $m_{HJ99}$ a drag coefficient of $C_d = 0.0025$ is used. The upper and lower bounds on the observationally constrained transfer coefficient $\Gamma_T$ are based on using $C_d = 0.0025$ and $C_d = 0.0097$ respectively. The bracketed melt rate estimates show the effect of setting the heat flux into the ice shelf $Q_i^T$ to zero.

| period | $T^*$ | $U_{7-19}$ | $m_{OBS}$ | $m_{MK18}$ | $m_{J10}$ | $m_{HJ99}$ | $\Gamma_T$ |
| --- | --- | --- | --- | --- | --- | --- | --- |
| | °C | cm s$^{-1}$ | m yr$^{-1}$ | m yr$^{-1}$ | m yr$^{-1}$ | m yr$^{-1}$ | $\times 10^{-3}$ |
| Feb–Nov | 0.20 | ∼ | 0.51 | 0.41 | ∼ | ∼ | ∼ |
| Apr–Jul | 0.23 | 4.2 | 0.62 | 0.46 | 2.35(2.49) | 1.35(1.44) | 2.7–5.3 |
| Aug–Nov | 0.16 | 3.9 | 0.30 | 0.29 | 1.44(1.52) | 0.83(0.89) | 2.1–4.2 |
| Apr–Nov | 0.20 | 4.0 | 0.46 | 0.37 | 1.85(1.96) | 1.07(1.13) | 2.5–5.0 |

$Q_i^T$=-0.5 W m$^{-2}$ (Fig. 9A). The ratio of heat lost to the ice ($Q_i^T$) to the latent heat flux ($Q_{latent}$) as a function of melt rate $m$ is constant and equal to 0.095 for $m > 0.2$ m yr$^{-1}$ (Fig. 9B), indicating that $\sim 10\%$ of the heat supplied by the ocean ($Q^T$) is lost to the ice. Based upon this result, $Q_T \sim 1.095 Q_{latent}$ at AM06.

## 4.3 Parameterized melt rates

We first investigate the different behaviours of the J10, HJ99 and MK18 parameterisations by considering the melt rates they
predict when applied to a short slice of observational data (Fig. 10). The shear-dependent parameterisations exhibit large

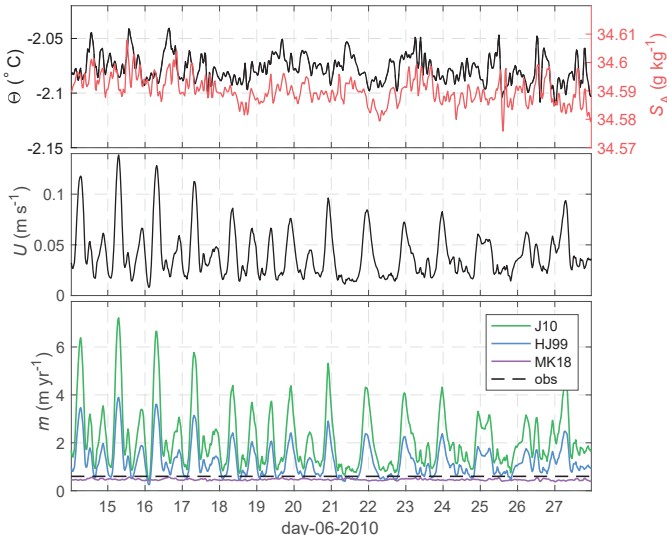

**Figure 10.** (top) $\Theta$ and $S_A$ from the upper microcat 4 m below the ice, (middle) current speed depth-averaged over 7–19 m below the ice base and (lower) parameterised melt rates for the three parameterisations compared in this study. While the J10 and HJ99 melt rates are strongly modulated by the tidal currents at AM06, MK18 varies only with temperature.

variability in melting on short timescales due to the tidal flow at AM06, which varies between 1 and 14 cm s$^{-1}$ over the two week period shown in Fig. 10. The convective parameterisation does not vary much on these timescales, as variability is driven solely by small amplitude temperature fluctuations. Both the J10 and HJ99 parameterisations predict much higher melt rates than are observed, while the MK18 convective parameterisation prediction is close to observations over this period.

The parameterisations are evaluated quantitatively by calculating the mean of the predicted melt rate over different averaging periods and comparing to the observed melt rate for the same period. The periods were chosen to minimise the uncertainty in the observed melt rates yet still capture some of the seasonal variability in the record. The observed $T^*$, $U$ and $m$ are presented in Table 3 alongside the predictions. The parameterisation that best fits the observations is the convective parameterisation based on the local slope angle, which is biased 20% low over period February–November. The shear-dependent parameterisations

(HJ99 and J10) are not evaluated over the full period February–November due to poor data return near the ice from the ADCP. J10 melt rates are 400% larger than the observed melt rates, while HJ99 melt rates are roughly 200% the observations. In all cases, the fit worsens by $\sim$6% if the heat flux into the ice is neglected. CTD profiles taken at the beginning of the observational period (Fig. 2) measure a temperature gradient of $d\Theta/dz = 0.0017\ °\mathrm{C\ m^{-1}}$ over the upper 10 m of the water column. Accordingly, if the melt rate calculated using J10 for $\Theta(d = 4)$ is 2 m yr$^{-1}$, taking $\Theta(d = 1)$ yields 1.95 m yr$^{-1}$ while

$\Theta(d = 10)$ results in 2.1 m yr$^{-1}$.

    The final column in Table 3 shows the best-fit transfer coefficient $\Gamma_T$ obtained by solving Eqs. A1, A2 and A3 given the observed melt rate and assuming $\Gamma_T/\Gamma_S = 35$. For $C_d = 0.0025$ we obtain $\Gamma_T \sim 5.0 \times 10^{-3}$ and for $C_d = 0.0097$ we obtain $\Gamma_T \sim 2.5 \times 10^{-3}$ over the period April–November.



### 4.4 Is it appropriate to apply a convective melting parameterisation at AM06?

Despite the relatively good agreement between observed melting and the MK18 parameterisation, it is not altogether clear that this parameterisation is appropriate for AM06 due to the presence of tidal currents. In laboratory experiments, McConnochie and Kerr (2017a) observe a transition from a convectively-controlled to a shear-controlled melting at a velocity of 2–4 cm s$^{-1}$ for $T^* \sim 0.5$ °C. The typical time-mean flow speed at AM06 ($U_{7-19}$) is $\sim 4.0$ cm s$^{-1}$, while instantaneous speeds can be in excess of 15 cm s$^{-1}$, suggesting that AM06 should be well within the shear-controlled melting regime. However, the effects 355 of a small slope angle and ambient stratification are not taken into account in this transition, which was determined for vertical ice.

In their study, Malyarenko et al. (2020) found that the convective melting parameterisation for a vertical ice face from Kerr and McConnochie (2015) captured observed melt rates well at sites on the Ross and Filchner-Ronne ice shelves during times where the Reynolds number for the diffusive sublayer ($Re_\delta$) was low, despite being applied to ice shelves with a shallow mean 360 slope. Based on these sites, they suggest a critical $Re_\delta$ for the convective-shear transition of $\sim 20$. However, as this threshold is based on the convective parameterisation for vertical (rather than sloping) ice, it is not clear that it can be applied to our site.

An important consideration for parameterising melt as a function of ice shelf slope in regional ocean models is the ice shelf basal topography. Here, we apply the MK18 parameterisation to the local slope measured by the ADCP, however, features such as our basal "scarp" (Fig. 7), or the basal terraces observed beneath Pine Island Glacier (Dutrieux et al., 2014), will be subgrid- 365 scale in the circumpolar or regional-scale ocean models for which ice-ocean parameterisations are needed. These models have resolution beneath ice shelves on the order of kilometers (e.g. Naughten et al., 2018b). As such, while the convective melt rate parameterisation is a relatively good fit for our observations, there remain significant challenges to the implementation of such a parameterisation in ocean models where small-scale variations in basal slope are not resolved.

### 4.5 Comparison with other direct melt rate measurements

Here we extend the comparison between observed and parameterised melt rates to include other published studies of ice shelf melt rate and *in situ* ocean observations from around Antarctica. Due to limitations in the data available this comparison is only made for the J10 parameterisation. The observed and parameterised melt rates, mean thermal forcing and current speed at each location are presented in Table 4, where the data are sourced from the relevant publications. More detail on the observational data is provided in the footnote of Table 4.

The ratio of observed to parameterised melt rates $m_{OBS}/m_{J10}$ is plotted as a function of the local thermal driving and flow speed in Fig. 11. The J10 parameterisation significantly over-predicts melt rates at many locations, particularly at warm and quiescent conditions. For example, beneath George VI Ice Shelf, where thermal driving is extremely high, predicted melt rates are $\sim 5000\%$ of the observed values ($m_{OBS}/m_{J10} = 0.02$). The exceptions to this are beneath the Filchner-Ronne Ice Shelf (FRIS), to which the transfer coefficients were tuned, and the Larsen C Ice Shelf (Davis and Nicholls, 2019). Both the FRIS 380 and Larsen C sites are characterised by low thermal driving ($T^* \sim 0.05$ °C) and strong, tidally-dominated flow.





There are many reasons why J10, a shear-dependent parameterisation, may not accurately reproduce melt rates, especially under warm and quiescent conditions. For example, another mechanism such as convection (Kerr and McConnochie, 2015; Mondal et al., 2019) or double-diffusive convection may be the dominant process driving melting. Stratification effects, a poorly constrained drag coefficient, or an inappropriate choice of input variables may also result in inaccurate melt rate predictions.

In high-resolution models, double-diffusive convection has been shown to drive melting beneath ice shelves under warm, low shear conditions (Middleton et al., 2021), forming a thermohaline staircase beneath the ice (Rosevear et al., 2021). Observations of a thermohaline staircase beneath George VI Ice Shelf (Kimura et al., 2015), which is subject to extremely high thermal driving, suggest that double-diffusive convection may drive melting there. Similarly, double-diffusive convection may play a role in melting at the quiescent grounding line of the Ross Ice Shelf (Begeman et al., 2018). Both these sites are characterised

by strong thermal forcing relative to the current speed.

For shear-dominated melting, surface buoyancy flux due to meltwater can inhibit vertical fluxes, decreasing the efficiency of heat and salt transfer (Vreugdenhil and Taylor, 2019). Thus, stratification effects may be responsible for the misfit between the parameterised and observed melt rates at AM06 and other sites. For example, a decrease in the efficiency of heat transport could explain the poor performance of the J10 parameterisation for the Amery (this study) and Ross (Stewart, 2018) ice shelf

sites which have similar current speeds to, but much higher thermal driving than, the Larsen C and Filchner-Ronne sites.

Another possible source of discrepancy between observed and parameterised melt rates is the drag coefficient. At AM06, a lack of information about the frictional properties of the ice base forces an arbitrary choice of $C_d$ in order to apply the shear-dependent parameterisations to the oceanographic data. This issue is not just specific to our study–in general, $C_d$ is extremely poorly constrained beneath ice shelves (e.g. Gwyther et al., 2015). Furthermore, $C_d$ is often used as a tuning parameter when

attempting to reconcile observed and parameterised melt rates (e.g. Jenkins et al., 2010b; Nicholls, 2018). In ice-ocean models, drag coefficients in the range 0.0015–0.003 are typically used (e.g. MacAyeal, 1984; Gwyther et al., 2015; Naughten et al., 2018b). Recent turbulence measurements beneath the Larsen C Ice Shelf were used to infer a drag coefficient of $C_d = 0.0022$ at a melting site with a cold, unstratified, tidally forced ISOBL (Davis and Nicholls, 2019). Beneath melting sea ice, values in the range 0.0025–0.01 have been measured (McPhee, 1992), while values in excess of 0.01 have been observed beneath sea ice

in the presence of rough platelet ice (Robinson et al., 2017). The drag coefficient $C_d = 0.0097$ recommended by Jenkins et al. (2010b) is within this range of estimates.

Poor agreement between the parameterised and observed melt rates may be a result of the depth at which the temperature, salinity and velocity are measured. At AM06 we observe that the boundary layer beneath the ice is stratified in both temperature and salinity, contrary to the paradigm of a well-mixed ISOBL on which the three-equation parameterisation is based. Other

borehole measurements such as those in the McMurdo (Robinson et al., 2010) and George VI (Kimura et al., 2015) ice shelves also show stratification in temperature and salinity below the ice. Near the calving front of the Ross Ice Shelf, the absence of a well-mixed layer beneath the ice was found to reduce the fit between the three-equation parameterisation and the observed melt rates (Stewart, 2018), and result in a sub-linear dependence of melt rate on temperature. The importance of measuring the current speed at an appropriate depth has also been demonstrated. Beneath the Larsen C Ice Shelf, Davis and Nicholls (2019)

found that at low flow speeds —when their fixed-depth velocity measurements were taken outside of the log layer—the drag

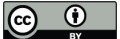



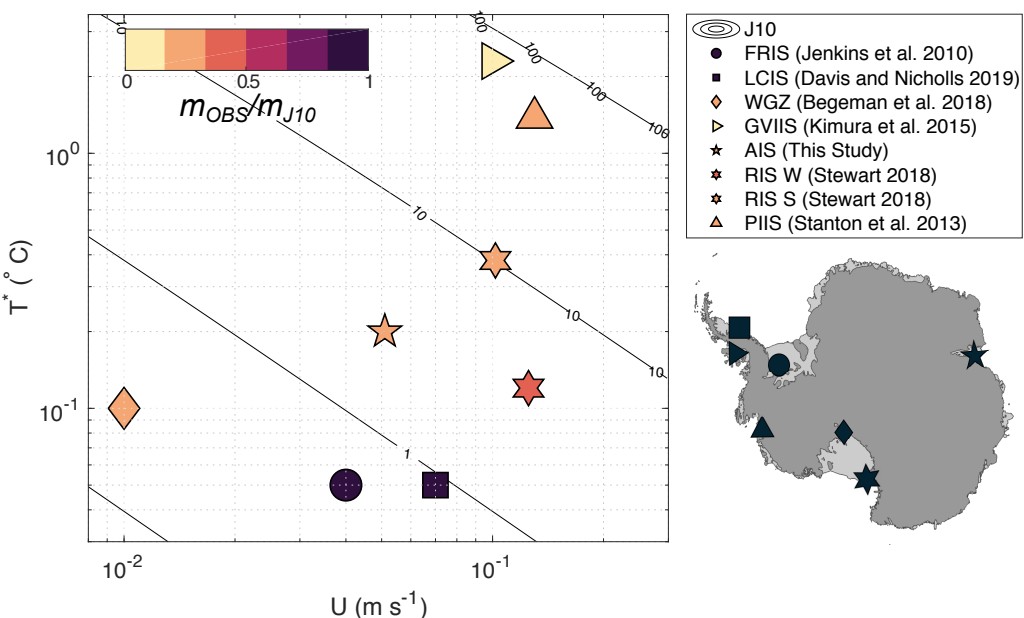

**Figure 11.** Ratio of observed ($m_{OBS}$) to predicted melt rate for the J10 parameterisation ($m_{J10}$) as a function of thermal driving ($T^*$) and free stream velocity ($U$) for published ice shelf datasets. Contours show $m_{J10}$. Map shows the location of the observations.

relationship (Eq. A4) did not estimate the friction velocity accurately, introducing large errors in the predicted melt rate. In the presence of steep basal topography, the problem of correctly identifying the depth at which $\Theta$, $S_A$ and $U$ should be sampled becomes even more challenging. For example, the basal scarp observed at AM06 may result in acceleration, stagnation or separation of the flow. Consequently $U_{7-19}$, which we use to predict melting, may not be representative of the flow speed next

to the ice. Finally, we highlight that these issues are not limited to observational studies. The numerical models for which these parameterisations were developed are also sensitive to the choice of sampling depth, as well as the way in which the meltwater flux is distributed (Gwyther et al., 2020). Sampling and distributing meltwater at the upper grid cell introduces a dependence of the melt rate on the model grid resolution.

## 5    Summary and Conclusions

In this paper we examined the relationship between basal melting and ocean conditions beneath the AIS using mooring data from a borehole drilled in 2010. The mooring location is characterised by the year-round presence of ISW derived from HSSW. The ISW is consistently warmer than the *in situ* freezing temperature, indicating it originates from shallower within the cavity, and the mean flow is oriented into the cavity at 220 °N. These observations are consistent with a "three dimensional" picture of cavity circulation with HSSW inflow on the eastern flank of the cavity being balanced by the outflow of ISW in the west. Basal

melting at the site is a modest $0.51\pm0.18$ m yr$^{-1}$ and varies seasonally with temperature and salinity. The warmest conditions and highest melt rates are observed in the Austral autumn and the coolest, lowest-melt conditions in the Austral spring. The




**Table 4.** Comparison between observed and predicted melt rates for a series of Antarctic ice shelves for which both melt rates and *in situ* oceanographic data are available. Observed variables presented are the time-mean current speed ($\bar{U}$), time-mean friction velocity ($u^*$), thermal driving ($\bar{T}^*$) and melt rate ($m_{OBS}$). Melt rate predictions made from J10 are also presented for each site. Where they are in bold, we have used the estimate from the original study. In the final column we note sites with a strong tidal component to the flow.

| location | period | $\bar{U}$ (cm s⁻¹) | $u^*$ (cm s⁻¹) | $\bar{T}^*$ (°C) | $m_{OBS}$ (m yr⁻¹) | $m_{J10}$ (m yr⁻¹) | Tides |
|---|---|---|---|---|---|---|---|
| AIS (this study) | Apr–Nov 2010 | 4.0 | | 0.20 | 0.46 | 1.9 | Y |
| FRIS (Jenkins et al., 2010b) | Jan–Dec 2001 | ~3.6[a] | | ~0.05[b] | 0.554±0.006 | **0.553** | Y |
| LCIS (Davis and Nicholls, 2019) | Dec 2011 – Feb 2013 | ~7.0[c] | 0.33[d] | ~0.05[e] | 0.7±1.0 | **0.69±0.6** | Y |
| George VI (Kimura et al., 2015) | Jan 2012 | 10.0 | | 2.3 | 1.4 | 71 | |
| RIS S (Stewart, 2018) I | Dec–Mar 2012–14 | 10.2 | | 0.38 | 2.7 | 10 | Y |
| RIS W(Stewart, 2018) II | Apr–Nov 2011–14 | 12.5 | | 0.12 | 1.4 | 3.8 | Y |
| PIIS (Stanton et al., 2013) | Dec 2012 | 13.0 | 0.86 | 1.39 | 14.6 | 50 | |
| WGZ (RIS) (Begeman et al., 2018) | Jan 2015 | 1.0 | | 0.1 | 0.05 | 0.24[f] | |

[a]Jenkins et al. (2010b) use a tidal model to infer current speeds in 2001, based on measurements made in 1996–1998 and the observation that tides dominated the currents at the site. The value used here is the mean from 1996–1998. Value should be considered approximate only.

[b]Value obtained from visual inspection of Fig. 4 of Jenkins et al. (2010b). Value should be considered approximate only.

[c]Calculated from Fig. 3 of Davis and Nicholls (2019) (upper instrument). Value should be considered approximate only.

[d]Calculated from $\bar{U}$ and observed drag coefficient $C_d = 0.0022$.

[e]Mixed layer temperature $T_{ML} = -2.06$-$2.04$ °C (from temperature profile in Fig. 2b or from text of Davis and Nicholls (2019)). Using $S_{ML} = 34.54$ psu and $p_i$=304 dbar, $T_f = -0.0573 S_{ML} + 0.0832 - 7.53 \times 10^{-4} p_i$ yields $T^* = 0.04$-$0.06$ °C. Value should be considered approximate only.

[f]This value differs from the reported value of 0.15 in Begeman et al. (2018). The discrepancy may be due to the inclusion of the conductive ice shelf heat flux by Begeman et al. (2018) or the use of a different drag coefficient other than the $C_d = 0.0097$ suggested in Jenkins et al. (2010b).




springtime minimum in melt is coincident with the most highly meltwater-modified conditions at AM06, as well as the weakest
residual flow. Tides dominate current variability, driving current speeds of $\sim$10 cm s$^{-1}$, while the superposition of the tidal
and mean flow can result in flow speeds in excess of $\sim$ 15.0 cm s$^{-1}$. A large scarp ($\sim$20 m in height) was discovered in the
underside of the ice shelf using the upwards-looking ADCP, adding to our growing understanding of the spatial complexity
of ice shelf bases. In addition, we have demonstrated the utility of an upwards looking ADCP for field studies of ice-shelf
ocean interactions. We were able to measure basal melt rates, ocean velocities and produce a map of the underside of the ice
base with a single instrument, demonstrating the advantage over a single-beam acoustic instrument, such as an upward looking
sonar (Stewart et al., 2019).

*In situ* oceanographic and melt rate observations were used to evaluate common ice-ocean parameterisations. Despite the
presence of tidal currents, we found that the convective, ice shelf-slope dependent parameterisation of McConnochie and Kerr
(2018) was the best-performing of the three parameterisations tested at AM06, underestimating observed melt rates by $\sim$20%.
The velocity-dependent parameterisations of Jenkins et al. (2010b) and Holland and Jenkins (1999) overestimated melting by
$400\%$ and $200\%$ respectively. Extension of our analysis to other published studies of *in situ* oceanographic data demonstrated
that the misfit between the Jenkins et al. (2010b) parameterisation and observations is widespread in temperature-velocity
space: the parameterisation only performs well under the coldest, most energetic conditions. Previous studies have shown that
this parameterisation performs poorly for warm and/or quiescent conditions. However, here we have shown that even cold
cavity ice shelves such as the Ross and Amery Ice Shelves, which have strong currents and only moderate (0.1–0.5 °C) thermal
driving, are not well represented by this parameterisation. Our results suggest that understanding the effects of buoyancy on
the ISOBL is a critical area for future studies aiming to improve parameterisations of basal melting in ocean-climate models.

*Data availability.*  The data are available at https://data.aad.gov.au/metadata/records/ASAC_1164_AM06

## Appendix A:  Basal melting parameterisations

### A1   Shear-controlled melting

The Holland and Jenkins (1999) and Jenkins et al. (2010a) parameterisations take the same general form. The the interface
temperature $T_b$ is assumed to be at freezing temperature at interface salinity and pressure $T_b = T_f(S_b, p_b)$, where interface
temperature and salinity are related by the linearised liquidus relationship:

$$T_b = \lambda_1 S_b + \lambda_2 + \lambda_3 p_b. \tag{A1}$$

Physical parameters $\lambda_1$, $\lambda_2$ & $\lambda_3$ are described in Table A1.

At the ice-ocean interface, the divergence of heat is balanced by a latent heat flux due to melting, with an equivalent bal-
ance for salt. The ice shelf melt rate ($m$) appears in the latent heat ($Q_{latent} = L_f m \rho_i$) and brine ($Q_{brine} = S_b m \rho_i$) fluxes





**Table A1.** Physical parameters used in melt parameterisation calculations

| name | symbol | unit | value |
|---|---|---|---|
| Thermal diffusivity ocean | $\kappa_T$ | m$^2$ s$^{-1}$ | $1.4 \times 10^{-7}$ |
| Thermal diffusivity ice shelf | $\kappa_{T,i}$ | m$^2$ s$^{-1}$ | $1.1 \times 10^{-6}$ |
| Salt diffusivity of ocean | $\kappa_S$ | m$^2$ s$^{-1}$ | $1.3 \times 10^{-9}$ |
| Latent heat fusion | $L_f$ | J kg$^{-1}$ | $3.34 \times 10^5$ |
| Specific heat capacity ice shelf | $c_i$ | J (kg K)$^{-1}$ | 2009 |
| Specific heat capacity ocean | $c_p$ | J (kg K)$^{-1}$ | 3974.0 |
| Ocean reference density | $\rho$ | kg m$^{-3}$ | 1030.0 |
| Ice shelf reference density | $\rho_i$ | kg m$^{-3}$ | 920.0 |
| Liquidus slope (salinity) | $\lambda_1$ | °C kg g$^{-1}$ | -0.069 |
| Liquidus slope (pressure) | $\lambda_2$ | °C dbar$^{-1}$ | $-7.6 \times 10^{-4}$ |
| Liquidus offset | $\lambda_3$ | °C | 0.0826 |
| Von-Karman's constant | $\kappa$ | $\sim$ | 0.41 |

respectively, where $L_f$ is the latent heat of freezing and $\rho_i$ is the ice density. The heat balance expression is given by:

$$\rho_i m L_f = \rho_i c_i \kappa_{T,i} \frac{\partial T_i}{\partial z}\bigg|_b - \rho c_p \Gamma_T u^* (T_b - T_{ML}) \tag{A2}$$

where the first term on the right hand side is the diffusive heat flux into the ice shelf, $(\partial T_i/\partial z)_b$ is the ice shelf vertical temperature gradient evaluated at the ice-ocean interface and $c_i$ and $\kappa_{T,i}$ are the heat capacity and thermal diffusivity of the ice

respectively. The second term on the right hand side is the oceanic heat flux, here parameterised in terms of the bulk temperature difference across the boundary layer ($T_b - T_{ML}$, where $T_{ML}$ denotes the mixed layer temperature), the friction velocity $u^*$ and a transfer coefficient $\Gamma_T$. Parameters $\rho$ and $c_p$ are the density and heat capacity of the ocean mixed layer respectively. An equivalent expression is given for the balance of salt:

$$\rho_i m (S_b - S_i) = \rho \Gamma_S u^* (S_b - S_{ML}), \tag{A3}$$

where $\Gamma_S$ is the salt transfer coefficient. In this expression the ice salinity and the diffusive salt flux within the ice are assumed to be zero. The friction velocity ($u^*$) is defined as the square root of the kinematic stress at the ice-ocean interface. However, in ocean models $u^*$ is typically estimated as a function of the free-stream current speed ($U$) through a simple parameterisation:

$$u^* = C_d^{1/2} U \tag{A4}$$

where drag coefficient $C_d$ is often taken to be 0.0025 (Gwyther et al., 2015).





### A1.1 Holland & Jenkins (1999): flow-dependent transfer coefficients

Holland and Jenkins (1999) use Eqs. A1–A3 with transfer coefficients ($\Gamma_T$, $\Gamma_S$) from the sea ice literature (McPhee et al., 1987):

$$\Gamma_T = \left[\frac{1}{\kappa}\ln\left(\frac{u^*\xi\eta^2}{fh_\nu}\right) + \frac{1}{2\xi\eta} + 12.5Pr^{2/3} - 8.5\right]^{-1} \tag{A5}$$

$$\Gamma_S = \left[\frac{1}{\kappa}\ln\left(\frac{u^*\xi\eta^2}{fh_\nu}\right) + \frac{1}{2\xi\eta} + 12.5Sc^{2/3} - 8.5\right]^{-1} \tag{A6}$$

where $Pr$ ($Sc$) is the Prandtl (Schmidt) number, $\kappa = 0.4$ is von Karman's constant, $f$ is the Coriolis parameter, $\xi = 0.052$ is a dimensionless constant, and $h_\nu = 5\nu/u^*$ the thickness of the viscous sublayer. The stability parameter ($\eta$) describes the influence of an interfacial buoyancy flux, which reduces the ISOBL depth. The buoyancy flux is itself determined by the melt rate. For $\eta = 1$, the parameterisation becomes analogous to that used by Jenkins (1991). Eqs. A2 and A3 are functions of $u^*$.

### A1.2 Jenkins *et. al.* (2010): constant transfer coefficients

Jenkins et al. (2010b) used ice shelf melting, upper-ocean temperature and current meter measurements to observationally constrain these transfer coefficients. As $u^*$ was not directly measured, they inverted for the products $\sqrt{C_d}\Gamma_T$ and $\sqrt{C_d}\Gamma_S$, which they term thermal and saline Stanton numbers, using Eq. A4 and assuming constant $C_d$. The best fit to the data was found for $\sqrt{C_d}\Gamma_T$=0.0011, $\sqrt{C_d}\Gamma_S$= 3.1×10$^{-5}$, assuming the ratio $\Gamma_T/\Gamma_S = 35$. Drawing on results from the sea ice literature (McPhee, 2008), they recommend the values $C_d = 0.0097$, $\Gamma_S$= 3.1×10$^{-4}$ and $\Gamma_T$=0.011.

### A2 McConnochie & Kerr (2018): convection-controlled melting

In the convective melting parameterisation of Kerr and McConnochie (2015), extended to sloping ice in McConnochie and Kerr (2018), the interface temperature is given by:

$$T_\infty - T_b = \frac{\rho_i L_f + \rho_i c_i (T_b - T_i)}{\rho c_p}\left(\frac{\kappa_S}{\kappa_T}\right)^{1/2}\left(\frac{S_\infty - S_b}{S_\infty - S_i}\right), \tag{A7}$$

where $\kappa_T$ and $\kappa_S$ are the molecular diffusivities of heat and salt and the subscript $\infty$ denotes the ambient ocean values. The melt rate is then given by:

$$m = \gamma\sin^{2/3}\theta\left(\frac{g(\rho_\infty - \rho_b)\kappa_s^2}{\rho_\infty\nu}\right)^{1/3}\left(\frac{S_\infty - S_b}{S_\infty - S_i}\right), \tag{A8}$$

where $\gamma$ is a constant equal to 0.09 (Kerr and McConnochie, 2015) and $\theta$ is the angle of the ice-ocean interface to the horizontal. Using the liquidus relationship (Eq. A1), this system of Eqs. can be solved for $T_b$, $S_b$ and $m$.



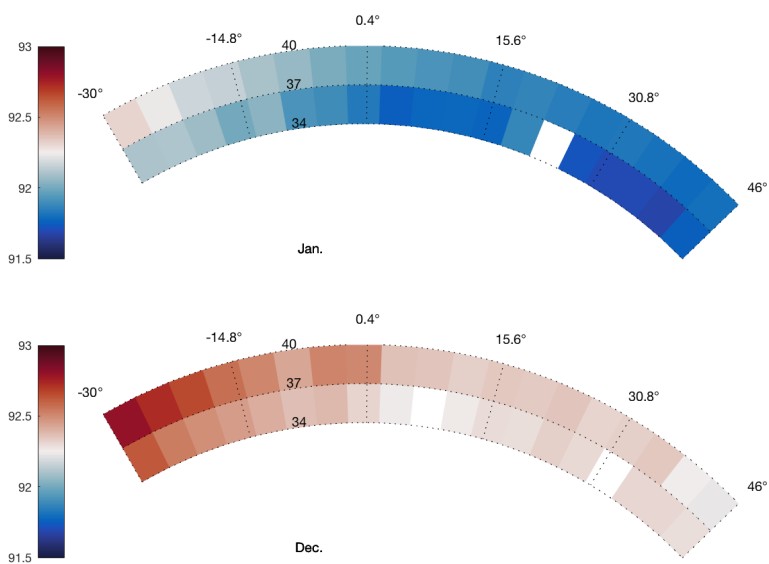

**Figure B1.** Range from the ADCP to the ice base in meters (colour) in January (upper panel) and December (lower panel) in polar co-ordinates.

**Appendix B**

*Author contributions.* B.G-F. and M.R. designed the research. M.R. performed the analysis. M.R., C.S. and B.G-F. contributed to interpretation of results and writing.

*Competing interests.* The authors declare that they have no competing interests.

*Acknowledgements.* We acknowledge logistic support from the Australian Antarctic Division and the many people who contributed to the
505 AMISOR project, Li Yuansheng (Polar Research Institute of China) for supplying the ADCP used in this study, Mark Rosenberg for data quality control and Rebecca Cowley for technical advice. This research was supported under the Australian Research Council's Special Research Initiative for Antarctic Gateway Partnership (Project ID SR140300001). Ben Galton-Fenzi received grant funding from the Australian Government as part of the Antarctic Science Collaboration Initiative program (ASCI000002). Craig Stevens is supported by the New Zealand Antarctic Science Platform ANTA1801.





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
