# Peer review of "Evaluation of basal melting parameterisations using *in situ* ocean and melting observations from the Amery Ice Shelf, East Antarctica"

_Ocean Science, 2021_

## Referee Comment (RC2)

Review of "Evaluation of basal melting parameterisations using in situ ocean and melting observations from the Amery Ice Shelf, East Antarctica" by Madeleine Rosevear, Benjamin Galton-Fenzi and Craig Stevens.

The authors present an outstanding set of ocean observations beneath the Amery ice shelf. The paper is clearly written, with a very good review in the Introduction and a thorough presentation of the new observations. This is a very useful manuscript as there are relatively few studies questioning the dominant use of shear-controlled formulations in ocean models. I have a single concern about the main conclusion of the paper on the overestimated melt rates in the shear-controlled formulations, as detailed below, but I think this can be addressed in a revised manuscript.

Major comment:

The values of $\sqrt{C_d}\Gamma_{T/S}$ provided by Jenkins et al. (2010) come from a fit to observations beneath the Ronne Ice Shelf. I don't think that these values should be considered as the best values to parameterize melt rates beneath Amery and I would expect these values to be recalculated for Amery based on the new observations. Similarly, the formulation of Holland and Jenkins (1999) seems to be based on a few days of measurements beneath sea ice in Greenland (Mc Phee 1987) and I would expect the authors to use the new observations beneath Amery to re-calibrate some of the parameters (e.g. $\xi$). In the way things are presented in sections 4.3 and 4.5 and in the Abstract and Conclusion, it is unclear to me whether the J10 and HJ99 parameterizations are highly biased because of a poor calibration or because their formulation is intrinsically wrong. With smaller values of $\sqrt{C_d}\Gamma_{T/S}$, there would be a better match for a majority of ice shelves. Or could the formulation be considered wrong because it requires significantly different $\sqrt{C_d}\Gamma_{T/S}$ values for the various Antarctic ice shelves? To be fair in the comparison between shear-controlled and convection-controlled formulations, would it be possible to do something like Table 4 but for the MK18 parameterization (at least for some of them or based on existing ice topography datasets)?

Minor comments and edits:

- L. 83: the second $\Gamma_T$ should be $\Gamma_S$

- L. 133: "used map" -> used to map.

- L. 319-320: "by surface accumulation" or ice convergence.

- L. 346-348: If I understand correctly, the last column of Table 3 is a recalibration of J10. Could you compare to the original values of J10? Could you do something similar to recalibrate HJ99 (e.g. changing $\xi$).

- Does the melt calculation in Table 4 and Figure 11 for the other locations take local tidal velocities into account? The formulation should not be based on $\overline{U}$ but more on something like $\sqrt{\overline{U^2}}$ (which could be roughly estimated from CATS or an equivalent tidal model).

---

## Author Comment (AC1)

OS-2021-111 response to R1 - Dr Carolyn Begeman

Dear Dr Begeman,

Thank you for taking the time to review our work, your constructive comments have been extremely useful in improving our manuscript.

In the text below we address all your comments item-by-item, where our responses are in blue.

We have made two small changes that were not specifically requested by either you or R2, but seemed natural when we made the requested changes:

- Firstly, thanks to R2's question about whether we use the mean+tidal velocities to calculate melt rates, we realized that we had misinterpreted the velocity reported in the text in Jenkins et. al., 2010. They state that the mean current value is 3.6 cm/s, which we misinterpreted as the time-mean of the current speed (i.e. the tidal+non-tidal components) rather than just the non-tidal component. Consequently, we don't know what mean(U) is, and cannot include the FRIS site on Fig. 10 any more. Since the parameterisation being tested Fig 10 (J10) is tuned to this dataset in the first place, this data point is not really needed in any case.
- Secondly, we modified the lengths of the averaging periods used to quantitatively assess the performance of the three parameterisation against observations from 4 months to 2-3 months. This facilitated the discussion that you requested around thermal driving, current speed and melting

Finally, we are pleased to acknowledge your contribution in the acknowledgements (text starting L554). Please let us know if this is not ok.

Major comments:
The main question that remains unaddressed is whether the authors could get a good fit for all three/four time periods with a single parameterization if c_d^1/2*Gamma_T were tuned. This is of great relevance to determining whether MK18 really is the best choice at this site.

Thanks for bringing this up. The short answer is no - we can't get a good fit for all four time periods. This is now discussed in section 4.3 (text starting L378): "The best-fit Stanton number also varies considerably between the different averaging periods. For example, we find $(C_d)^{1/2}\Gamma_T$=0.00017 for the July--September period, while $(C_d)^{1/2}\Gamma_T$=0.00031 for April-June. The fact that one value cannot be used year-round suggests that the functional form of J10 is not appropriate for all AM06 conditions."

I'd also like to see some discussion of the relationships between thermal driving and melt rate (and u_mean, u_tidal and melt rate) for each of the four periods analyzed. I imagine there is some reason the authors have omitted this discussion (nor plotted dT vs. melt rate), but this reason should be stated.

Thanks for this comment. The reason we have not plotted m vs T* or U is that we don't observe a systematic variation of melting with either variable. We have added discussion of this just before we compare the observed and parameterised melt rates

(section 4.3 starting L355). "... we briefly describe the relationship between the observed melt rate ($m$) and ocean forcing ($T^*$, $U$) for the 2-3 month averaging periods (Table 3). As expected, the lowest melt period (July-September) coincides with the weakest currents and cool temperatures, while high melting coincides with warmer temperatures and faster current speeds (e.g. April-June). However, while melting is nearly three times higher in the April-June period than the July-September period, $T^*$ and $U$ are only 16% and 35% larger respectively. Similarly, melting in the October-November period is nearly double that of the July-September period, despite extremely similar ocean conditions during the two periods."

I would also have appreciated more contextualization of the melt seasonality with, e.g., seasonal variability in AASW and HSSW properties and whether the phase relationships are compatible with advection timescales from the ice-shelf front to the observational site. I see that you cite ISW concentrations but do you have a hypothesis that would explain the timing of those changes?

For current speeds in the range of 2-5cm/s, the advection timescale from the ice shelf front (150km away) is in the range 35-85 days. However, previous studies suggest eddying flow (Herriaz-Borreguerro et. al., 2015) with meandering currents and recirculating features (Galton-Fenzi et. al., 2012) beneath the AIS, which would be expected to increase the advection times from the ice shelf front, and introduce the potential for "decoupling" the water mass properties inside and outside of the cavity. Because water mass properties at AM06 are quite well constrained to the meltwater mixing-line year-round, we hypothesize that the variation that we see is driven by increased residence time of the water beneath the ice shelf, allowing a higher degree of meltwater modification.

However, to determine the dynamics responsible for this, it would have been necessary to draw in data from other boreholes and from moorings outside the cavity. Ultimately, we decided that this would broaden the scope of the paper too much, and distract from the focus on evaluating melting parameterisations.  It certainly provides a good guide for future study.

Specific comments:

L5: Later, explain how your application of the ADCP is novel

We have clarified this at the start of section 2.1

L34: I'm not sure what exactly you mean by intermediate depths, but I'd say that for warm water cavities mCDW can also drive melting at deep depths.

We agree and have removed the reference to intermediate depths.

L36: This section heading was unexpected. Attempt to offer transition at the end of the previous paragraph.

This is a fair comment. We have added the following text to help transition: CDW-dominated cavities are often termed "warm cavities", while cavities dominated by HSSW and AASW are known as "cold cavities". The three largest Antarctic ice shelves -Ross, Filchner-Ronne and Amery- are all cold-cavity ice shelves.

[L219] "During this period... " implies that U_7-19 > U_19-91 from Aug-Dec but this appears to only be the case from Oct-Nov

   We agree and have amended the text.

L126: more details about the mooring needed, particularly given that the ADCP swings. By how much is the ADCP and other sensors moving? Is the mooring end affixed to the seafloor? How might this motion contaminate the measurements? How have you accounted for this motion?

Thanks for this comment, we agree that it is important to include this info. Weights were affixed to the end of the mooring to tension the cables and minimize motion (L133).This was quite effective as observed using the pressure sensor of the middle microcat, 40m below the ADCP, indicating that movement of the ADCP was minimal. The maximum pressure anomaly was -0.9 dbar, associated with a vertical excursion of <1m, however, excursions were typically much smaller than this. In order to avoid contamination of the range and melt rate measurements by this motion, we have excluded data when the pressure anomaly at the middle microcat is<-0.25. However, since excursions were small, these criteria results in only 2% data loss. We have added text to this effect at the end of Section 2.2 (starting L162).

Section 2.1: I found the relationship between the morphology determination and the melt rate determination confusing here. I understand the space and time bins you've used for the melt rate determination but not those for the basal morphology shown in Figure 7. In addition, since you say the BT is noisy, what is the uncertainty on the morphology shown in Figure 7?

   Thanks for this comment. We have clarified this in the caption of Fig. 7. For this application we use 1x1m bins, and average the full year of data to get the interface depth for each bin (since the meltrate (0.5m/yr) is much smaller than the variation in depth (20m) we do not account for this). The typical standard deviation of each bin is 1.3m.

L148: It's unclear how this heading range is chosen. Is this the range that is closer than 100m from the sensor?

   Thanks. It is the intersection between the region that is closer than 100m from the sensor and region/s where we had good data return. In practice, some of the ice base was sampled too infrequently to calculate melt rates. We have clarified this in the text (starting L157)

Table 1: Why doesn't interface pressure have a start date? Isn't this measured at the time of the CTD casts?

   Good point, it is. This has been added.

L171: Is this slope compatible with the eddy diffusivity of heat being greater than salt? If so, state that.

Great catch - it isn't (which we had not realized). The text now reflects this, and briefly discusses a mechanism that could steepen the slope: diffusive convection. (starting L187).

L190: "This indicates..." Logic of this sentence is unclear. I think you're talking about meltwater accumulating as mean flow is roughly from AM02 to AM06. Please clarify. In general, your explanation of why freshening occurs in spring wasn't clear to me.

Thanks, we have rephrased this to clarify this point.

Figure 5. Is the grey an uncertainty envelope? Explain in the caption.

Done

Figure 6. I assume the current direction is measured from north? Specify in caption. In addition, You don't explain why the RMS data is missing until later. Please state in the caption or when Figure 6 is first referenced.

Thanks, done.

L205: "consistently biased" Have you ruled out instrumental bias? If so, please state in the manuscript.

We are more inclined to trust our ADCP measurements over the CATS tidal model, given that tide modelling is sensitive to bathymetry, which is poorly constrained in general beneath Antarctic ice shelves. We have added some text to this effect, starting L226.

Figure 8: Why are the horizontal lines 2 months long if the measurements are at monthly intervals?

The lines are 2 months long because We used centered differencing to calculate the melt rate from the interface depth measurements. We have clarified this in the figure caption.

Section 3.5: Is there a reason why you don't have a panel similar to those in figure B1 but with the annual average melt rate? Can you identify significant spatial variation in melt rate? If not, why not?

The annual average melt rate (fig. below) has not been included because we do not see significant spatial variation in melt rate.

[Figure]

Figure: Spatial distribution of melt rate (m/yr), annual average.

Section 4.1: It would be helpful to include in this section the explanation for why you average melt rates over 3 months.

Thanks we have added this in section 4.1 (starting L307).

Section 4.4: Readers should be reminded that MK18 does not depend on current speed (residual or tidal)

Done. (text starting L389)

Table 3: I'd like to see the melt rate uncertainty over these 3 month windows

Done.

L289: Can you remind us what the upper bound on the microcat depth is, here or on the lines when you discuss sensitivity to this choice? Can you also indicate to us that you are going to discuss sensitivity to depth later?

The upper bound has been added.The sensitivity to depth is discussed in the paragraph immediately following this.

L305: Can you include a figure in the supplement or appendix that shows us that the upper water column velocity structure is relatively depth independent?

This is a good idea, thank you. We now include a new figure which shows this qualitatively (holmoller plot showing the current speed in depth and time) and quantitatively (histogram of the ratio between velocity in the top bin and the 4-bin mean).

[Figure]

L307: This missing data should be mentioned earlier.

    Thanks, this is now mentioned in section 3.2 when the current data is first presented.

Figure 9: I really appreciate that the authors have computed the conductive flux, as most studies neglect this term. However, I think that this figure would be more appropriate for the appendix given the relative importance of the conductive heat flux sensitivity to melt rate when compared with the other figures and results.

    We have moved Fig. 9 to the appendix.

L335: Here, it's unclear whether "different averaging periods" refers to averaging periods of different durations or different start/end times.

    This text has been removed as the averaging periods are now introduced at the start of section 4.1

L336: "The periods were…" I would have rather had this information when you were introducing data choices.

Addressed above.

L350: Remind readers that MK18 is a convective parameterization

Done.

L361: This is confusing: Malyrenko et al. suggest a critical Re_delta for horizontal ice settings yet "it's not clear that it can be applied to our (horizontal) site"

We have now calculated Re_delta for the AM06 data and compared it to the threshold from Malyarenko et al. Starting L399.

L362: In this paragraph, it would be good to get a sense for how much getting the local slope right might matter. That is, how much would the predicted melt rate change if the large-scale slope of 0.1deg were used instead of the local 9deg slope?

The melting is very sensitive to the slope angle - new melt rate would be ~3cm/yr. We have added text to this effect starting L406.

L427: Here or elsewhere, clarify that what you hypothesize is a mostly barotropic flow bringing both ISW and HSSW from shallower depths near the ice-shelf front to your observational site.

Done. Starting L473: "The mean flow is oriented into the cavity at 220 N and exhibits little vertical shear over the upper 100 m of the water column. We hypothesize a mainly-barotropic flow advecting HSSW from the calving front, which is modified by the addition of meltwater to become ISW as it travels beneath the ice shelf and past AM06. This is consistent with a "three dimensional"' picture …"

Technical comments:

All the below changes have been made

L1: Phrasing here is convoluted. Would be clearer to write "is accelerating loss of grounded ice" or similar.

L9: Might as well express the seasonal variability as a percentage of the mean here

L11: Can remove ~ since you've already specified "typical" speeds

Figure 1: Specify what dashed line is. Schematic arrow for the sub-ice-shelf circulation would also be helpful.

L23: demonstrated >> demonstrated that

L27: melts >> melt

L30,31: watermasses >> water masses

L43: The ISW >> ISW

L66: In the >> The

L71: roughness >> ice shelf basal roughness

L74: parameterised >> parameterised in these ocean models

L83: Gamma_T >> Gamma_S

L92: Use T_b-T_ML instead of T' or specify what T' refers to oce

L105: and with >> and scales as

L110: the tendency of >> found that … tended to reproduce

L113: the questions >> these questions

L118: ice shelf >> ice shelf there

L148: Can you choose a different notation for heading or ice shelf slope so that they cannot be confused?Figure 7: Make explicit that the black box is the -30 < theta < 46 in caption or text.

L235: Remind us that AM06 is on the eastern flank

L248: "The maximum…" The logic of this sentence is unclear.

L296: I'm not sure why you use the LOW abbreviation as I don't think it appears again.

L327: Since the equations are in the appendix, the readers need to be reminded what the variables refer to if it's been a while since you introduced them. Here, remind them what Q_T is.

L436: bases >> basal topography

L454: delete "the"

---

## Author Comment (AC2)

OS-2021-111 response to R2

Dear Reviewer,

Thank you for taking the time to review our work, your constructive comments have been extremely useful in improving our manuscript.

In the text below we address all your comments item-by-item, where our responses are in blue.

We have made two small changes that were not specifically requested by either you or R1, but seemed a logical extension when we made the requested changes:

- Firstly, thanks to your question about whether we use the mean+tidal velocities to calculate melt rates, we realized that we had misinterpreted the velocity reported in the text in Jenkins et. al., 2010. They state that the mean current value is 3.6 cm/s, which we misinterpreted as the time-mean of the current speed (i.e. the tidal+non-tidal components) rather than just the non-tidal component. Consequently, we don't know what mean(U) is, and cannot include the FRIS site on Fig. 10 any more. Since the parameterisation being tested Fig 10 (J10) is tuned to this dataset in the first place, this data point is not really needed in any case.
- Secondly, we modified the lengths of the averaging periods used to quantitatively assess the performance of the three parameterisation against observations from 4 months to 2-3 months. This facilitated the discussion that R1 requested around thermal driving, current speed and melting

Finally, we wish to acknowledge your contribution (text starting L554).

Major comment:
The values of $C_d\Gamma_{T/S}$ provided by Jenkins et al. (2010) come from a fit to observations beneath the Ronne Ice Shelf. I don't think that these values should be considered as the best values to parameterize melt rates beneath Amery and I would expect these values to be recalculated for Amery based on the new observations. Similarly, the formulation of Holland and Jenkins (1999) seems to be based on a few days of measurements beneath sea ice in Greenland (Mc Phee 1987) and I would expect the authors to use the new observations beneath Amery to re-calibrate some of the parameters (e.g. $\xi$).

      Regarding the Jenkins 2010 parameterisation, one of the main aims of the paper was to determine whether parameters fit to the Ronne ice shelf could be used for the Amery (and others). We found that they could not, indicating that a parameterisation of this type (($C_d)^{1/2}\Gamma_{T/S}$ = constant) will be inaccurate if applied everywhere. We think it is worthwhile to have established this. We have presented the best fit ($C_d)^{1/2}\Gamma_{T/S}$ values for the Amery as a point of comparison, with new discussion starting L380.

We have not recalibrated other parameters using AM06 data for a few reasons. Firstly, since turbulent quantities were not measured, the system is undetermined. If $u^*$ were measured (as in Davis et al 2019), then $C_d$ would be known and we could start to look at the transfer coefficients in more detail. Secondly, since the melt rate data has very poor temporal resolution, it is challenging to even test the functional form of the parameterisations (since quantities such as $U$, $T^*$ and $m$ do not vary much when averaged over the long periods we found to be necessary). Thirdly, we wanted to move away from the approach of tuning parameterisations to each ice

shelf, since if a parameterisation needs to be tuned for each ice shelf, or each site, it is not particularly useful for modeling purposes.

In light of this, our approach was to consider which parameterisations could best replicate observed melting at AM06. Where the parameterisations could not replicate the melting, we investigated how the ocean conditions might explain the poor performance (e.g. section 4.4 and section 4.5 from L425 onwards)

In the way things are presented in sections 4.3 and 4.5 and in the Abstract and Conclusion, it is unclear to me whether the J10 and HJ99 parameterizations are highly biased because of a poor calibration or because their formulation is intrinsically wrong. With smaller values of $C_d\Gamma_{T/S}$, there would be a better match for a majority of ice shelves. Or could the formulation be considered wrong because it requires significantly different $C_d\Gamma_{T/S}$ values for the various Antarctic ice shelves?

Thanks very much for pointing out that this isn't clear in the paper at present. For J10, we propose that the formulation is intrinsically wrong, or at least not applicable over the full range of relevant/observed ice shelf conditions. We have now clarified this in section 4.5 (starting L423) and reiterated it in the Conclusions (starting L494).

To be fair in the comparison between shearcontrolled and convection-controlled formulations, would it be possible to do something like Table 4 but for the MK18 parameterization (at least for some of them or based on existing ice topography datasets)?

While this is possible, it would not be meaningful since existing ice topography datasets are too coarsely resolved/"smooth" to capture the local slope, like what we measure at AM06. (see discussion L402). Without this, a convective parameterization will significantly underestimate melting. For example, if we use a basal slope of theta=0.2 degrees rather than ~9 degrees at the AM06 site, the MK18 parameterisation predicts a melt rate of ~3 cm/yr.

Minor comments and edits:
L. 83: the second $\Gamma T$ should be $\Gamma S$
thanks, done

L. 133: "used map" -> used to map.
thanks, done

L. 319-320: "by surface accumulation" or ice convergence.
thanks, done

L. 346-348: If I understand correctly, the last column of Table 3 is a recalibration of J10. Could you compare to the original values of J10?
Thanks, we have added some discussion about this starting L379

Could you do something similar to recalibrate HJ99 (e.g. changing $\xi$).
See response to major comments.

Does the melt calculation in Table 4 and Figure 11 for the other locations take local tidal velocities into account? The formulation should not be based on $U$ but more on something like $U!$ (which could be roughly estimated from CATS or an equivalent tidal model).

In summary, yes, the melt calculation does take the tidal velocity into account, since we use the measured velocity at each site which is a combination of the tidal and mean components. We use the time-mean current speed rather than the background current speed plus the root-mean-square tidal velocity, since the latter approach was suggested as a way to include tidal effects on melting in ocean models that do not include tides.

---

## Author Response (AR2)

Dear Editor and Reviewers,

Thank you very much for giving your time to evaluate/handle our manuscript. We really appreciated the clear communication, timely reviews and constructive feedback.

We have made all the minor corrections recommended by Dr Carolyn Begeman, except:
*L211: Isn't the reason more that mCDW intrusions are spatially localized and that AM02 and AM06 aren't quite along the same flowpath than that it is due to a deeper ice draft?*
Since —while plausible— this is speculation and not something that we can test with the available data.

Thank you again,
Madelaine Rosevear